# Entropy Measures for Transition Matrices in Random Systems

Zhaohui Chen[1], René Meyer[1], and Zhuo-Yu Xian[*1,2]

[1]Institute for Theoretical Physics and Astrophysics and
Würzburg-Dresden Cluster of Excellence ct.qmat,
Julius-Maximilians-Universität Würzburg, 97074 Würzburg, Germany
[2]Department of Physics, Freie Universität Berlin, Arnimallee 14,
DE-14195 Berlin, Germany

## Abstract

A transition matrix can be constructed through the partial contraction of two given quantum states. We analyze and compare four different definitions of entropy for transition matrices, including (modified) pseudo entropy, SVD entropy, and ABB entropy. We examine the probabilistic interpretation of each entropy measure and show that only the distillation interpretation of ABB entropy corresponds to the joint success probability of distilling entanglement between the two quantum states used to construct the transition matrix. Combining the transition matrix with preceding measurements and subsequent non-unitary operations, the ABB entropy either decreases or remains unchanged, whereas the pseudo-entropy and SVD entropy may increase or decrease. We further apply these entropy measures to transition matrices constructed from several ensembles: (i) pairs of independent Haar-random states; (ii) bi-orthogonal eigenstates of non-Hermitian random systems; and (iii) bi-orthogonal states in $PT$-symmetric systems near their exceptional points. Across all cases considered, the SVD and ABB entropies of the transition matrix closely mirror the behavior of the subsystem entanglement entropy of a single random state, in contrast to the (modified) pseudo entropy, which can exceed the bound of subsystem size, fail to scale with system size, or even take complex values.

*Corresponding author: zhuo-yu.xian@fu-berlin.de

# 1 Introduction

Quantum teleportation [1, 2] is a foundational protocol in quantum information theory that enables the transfer of quantum states between two distant systems without direct physical transmission. It relies on a shared maximally entangled state (MES) $|\psi_1\rangle_{bc}$ between Alice (holding system $b$) and Bob (holding system $c$), along with classical communication.

By performing a joint measurement on Alice's system $b$ and $a$ in the maximally entangled basis, where $a$ carries an unknown input state $|\phi\rangle_a$, Bob's system $c$ collapses into the desired state $|\phi'\rangle_c$ up to a known and operable unitary transformation determined by the outcome of the measurement.

As illustrated in Fig. 1, for an entangled state $|\psi_1\rangle_{bc}$, if Alice applies a joint post-selection (projection) of a specific entangled state $|\psi_2\rangle_{ab}$ on her system $b$ and $a$ rather than a joint measurement, the transition of the unknown state $|\phi\rangle_a$ is described by applying

a reduced transfer matrix $\tau$ on the unknown state $|\phi\rangle_a$. Post-selection was once used to construct the time-symmetrical ensemble [3] and also applied in weak measurement [4, 5]. Additionally, post-selection has been proposed as a method to reconcile black hole evaporation with unitarity by imposing a final state projection at the black hole singularity [6–9].

Unlike perfect teleportation, which transfers information exactly and without loss, a post-selection transition matrix $\tau$ conveys only partial information of $|\phi\rangle_a$ from the system $a$ to $c$. In general teleportation that involves completely positive and trace-preserving (CPTP) quantum channels, the fidelity is typically applied to quantify the deviation between the input and output states [10–15]. However, in our context, the transition matrix $\tau$ is not trace-preserving any longer and hence does not constitute a quantum channel, so we set aside the fidelity prescription and instead try to employ several notions of entropy to quantify how much information could be transferred through this transition matrix $\tau$.

The first measure is the pseudo entropy [16], defined as the von Neumann entropy of the normalized transition matrix $\hat{\tau}$. It originates from the holographic dual of the minimal surface area on a time-dependent asymptotically AdS spacetime, serving as a generalization of holographic entanglement entropy [17–21]. Since its introduction, pseudo entropy has been extensively studied in various contexts, including quantum spin models and free scalar field theories [22, 23], quantum field theories and CFT [9, 24–35]. In the holographic setting, pseudo entropy has been calculated in AdS/BCFT, and its phase transition structure has been quantitatively analyzed in [36, 37]. A related development is the thermal pseudo entropy, which generalizes thermal entropy to complex inverse temperatures and has been studied in several setups [38]. Furthermore, both timelike entanglement entropy and complex-valued holographic entanglement entropy in dS/CFT have been shown to be realizations of pseudo entropy [39–44]. Pseudo entropy has also found some applications beyond being a entanglement measure, for instance, a subsystem generalization of the spectral form factor (SFF) [45, 46] can be introduced via pseudo entropy [47], with further connections to the SFF explored in SYK models [48].

In non-Hermitian systems, bi-orthogonal quantum mechanics has been developed to preserve key properties such as orthogonality and the probabilistic interpretation [49]. The von Neumann entropy of the normalized transition matrix $\hat{\tau}$ constructed from the bi-orthogonal basis, essentially a type of pseudo entropy, was initially studied in the $PT$-symmetric SSH model at criticality [50]. There, the entropy displays a logarithmic scaling with negative central charges, indicating that this model is governed by a non-unitary CFT. Further studies on pseudo entropy based on the bi-orthogonal bases have been carried out in [51–55].

Recently, a modified version of pseudo entropy was introduced, defined by taking the logarithm of absolute eigenvalues of $\hat{\tau}$ [56]. This modified pseudo entropy has shown effective in measuring the correct negative central charges in certain critical non-Hermitian systems such as the two-legged SSH model or the $q$-deformed XXZ model [50, 57]. Moreover, it successfully eliminates nonphysical singularities that arise in Rényi pseudo entropies, as demonstrated in the AKLT model. Since its proposal, the modified pseudo entropy has been further applied to non-Hermitian quantum systems [58–62], non-Hermitian random matrix model [63], and also to defining quantum mutual information in time [64,65]. The interpretations of pseudo entropy and its modifications based on the correlation matrices in non-Hermitian free-fermion systems were reviewed in [66].

However, there are some serious issues associated with (modified) pseudo entropy. Due

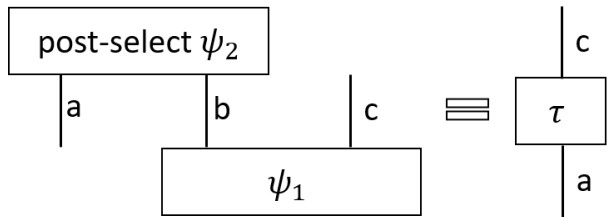

Figure 1: Given an entangled state $|\psi_1\rangle_{bc}$ shared by $b$ and $c$, the post-selection is performed on the joint system $ab$ with one specific entangled state $|\psi_2\rangle_{ab}$. This operation amounts to transferring the information of an unknown state from $a$ to $c$ via the transition matrix $\tau$.

to the non-hermiticity of the transition matrix $\hat{\tau}$, (modified) pseudo entropy often takes negative or complex values instead of being real and positive, and, in some cases, can even diverge. For instance, in $PT$-symmetric non-Hermitian system [50, 56], (modified) pseudo entropy can become greatly enhanced near exceptional points, across which a $PT$-symmetric quantum state transitions into a $PT$-broken one, and ultimately diverges at such exceptional points, exceeding the logarithm of the Hilbert space dimension. Moreover, as opposed to entanglement entropy, it has been shown [16] that (modified) pseudo entropy does not necessarily satisfy the subadditivity and strong subadditivity, even for transition matrices with real and positive spectra.

To address issues related to complex values and divergences of the (modified) pseudo entropy, the SVD entropy was introduced in [67]. This definition relies on the singular value decomposition (SVD) of the transition matrix $\tau$ to construct a density matrix $\bar{\tau}$. This entropy is real, positive, and strictly bounded by the logarithm of Hilbert space dimension. The properties of SVD entropy were further explored in various contexts, including general and holographic CFTs, and Chern-Simons theory [67]. A related development is the notion of excess in the SVD entropy, introduced as a useful measure to quantify the difference between pre-selected and post-selected states [68]. In spite of this, some new issues also arise, for example the difficulty of computing it in field theory due to the presence of the square roots in the definition of the SVD entropy. Additionally, the standard entropy inequalities such as subadditivity and strong subadditivity do not hold any longer for the SVD entropy.

Probabilistic interpretations of pseudo entropy and SVD entropy were presented in [16, 67], building upon the framework of entanglement distillation [69]. In our work, we will elaborate that the distillation of large-copy transition matrices for these two entropies does not possess a genuine probabilistic interpretation within the standard formalism of quantum mechanics. Given this, we introduce a novel entropy measure to quantify the transition amount based on a real physical process. The entropy, termed the ABB entropy previously discussed in [70], is defined as the von Neumann entropy of $\tilde{\tau}\tilde{\tau}^\dagger$, where $\tilde{\tau}$ denotes the state-normalized form of $\tau$. To gain further insight, we explore the probabilistic interpretation of ABB entropy based on two distillation methods in arbitrary dimension and compare it with those for the (modified) pseudo entropy and SVD entropy. In this paper, we choose the notations for transition matrices of different normalizations as shown in Tab. 1.

In this paper, our primary objective is to analyze and compare the properties of these four different entropy measures from various perspectives, as well as probabilistic interpretation. It is well known that entanglement entropy cannot increase under local operations and classical communication (LOCC) [71]. Building on this principle, we

| Transition matrices | $\hat{\tau}$ | $\bar{\tau}$ | $\tilde{\tau}$ |
|---|---|---|---|
| Normalizations | $\dfrac{\tau}{\operatorname{Tr}\tau}$ | $\dfrac{\sqrt{\tau^{\dagger}\tau}}{\operatorname{Tr}\sqrt{\tau^{\dagger}\tau}}$ | $\dfrac{\tau}{\sqrt{\operatorname{Tr}\left[\tau\tau^{\dagger}\right]}}$ |
| Schmidt coefficients | $z_i$ | $q_i$ | $\sqrt{p_i}$ |
| Spectra for entropy | $\lambda_i$ | $q_i$ | $p_i\left(\tilde{\tau}\tilde{\tau}^{\dagger}\right)$ |

Table 1: The notation of different normalizations of a given transition matrix $\tau$ for (modified) pseudo entropy, SVD entropy and ABB entropy.

firstly investigate how these four entropy measures respond to generalized measurement and successive quantum operation applied to the transition matrix $\tau$, aiming to identify those that remain non-increasing under such transformations.

The Page curve refers to the entanglement entropy as a function of the logarithm of the subsystem's Hilbert space dimension in the ensemble average. Its asymptotic result in Haar random ensemble was presented in [72, 73], and it was extensively studied in black hole information paradox [74, 75]. The probability distribution of the pseudo entropy for transition matrices constructed from two independent Haar random states was given in [16], and the corresponding ensemble-averaged SVD entropy was presented in [67], which displays an explicit scaling behavior with subsystem size, analogous to the Page curve. Additionally, the ensemble-averaged modified pseudo entropy was calculated numerically in non-Hermitian chaotic systems [63], where transition matrices are constructed from bi-orthogonal eigenstates, and no explicit scaling behavior appears; instead the entropy is significantly suppressed. In this work, we will also analyze and compare the behavior of all four entropy measures in the ensemble average for both two independent Haar random states and two correlated random states in non-Hermitian chaotic systems.

Lastly, we apply these four entropy measures to strongly entangled states in a strongly coupled quantum mechanical model, the SYK Lindbladian with linear jump operators [76]. This SYK Lindbladian is $PT$ symmetric, so we will investigate the behavior of four entropy measures in the $PT$-symmetric and $PT$-broken regions, especially the regions near exceptional points.

The remainder of this paper is organized as follows. In Sec. 2, we introduce the ABB entropy, review the pseudo entropy, modified pseudo entropy, and SVD entropy, and analyze the behavior of the different entropies when the transition matrix undergoes generalized measurements and successive quantum operations. In Sec. 3, we discuss the probabilistic interpretations of all four entropy measures following the perspective of entanglement distillation. In Secs. 4 and 5, we calculate the ensemble averages of each entropy in the contexts of two Haar random states and non-Hermitian chaotic systems, respectively. In Sec. 6, we analyze the behavior of four entropies in $PT$-symmetric SYK Lindbladian. Finally, in Sec. 7, we provide concluding remarks and future outlook.

# 2 Various entropy measures for transition matrices

## 2.1 Transition matrices from post-selection

As already mentioned in the introduction, we construct the transition matrix $\tau$ through post-selection. Consider a scenario in which Alice and Bob share an entangled state $|\psi_1\rangle_{bc}$ across their systems $b$ and $c$. To transfer the information of an unknown quantum state

$|\phi\rangle_a$ to Bob, Alice performs the post-selection of a specific entangled state $|\psi_2\rangle_{ab}$ on the joint system $ab$. As illustrated in Fig. 1, the post-selection is described by the transition matrix $\tau$ as

$$\tau = \mathrm{Tr}_b \, |\psi_1\rangle_{bc} \, \langle\psi_2|_{ab} : \mathcal{H}_a \to \mathcal{H}_c \,, \tag{1}$$

which is a linear map from the Hilbert space of system $a$ to that of $c$. The resulting output state is given up to normalization by

$$\mathrm{Tr}_{ab} \, |\phi\rangle_a \, |\psi_1\rangle_{bc} \, \langle\psi_2|_{ab} = \tau \, |\phi\rangle_a \,, \tag{2}$$

with the post-selection probability $(\langle\phi|_a \, \langle\psi_1|_{bc}) \, |\psi_2\rangle_{ab} \, \langle\psi_2|_{ab} \, (|\phi\rangle_a \, |\psi_1\rangle_{bc}) = \langle\phi|_a \, \tau^\dagger \tau \, |\phi\rangle_a$, where $\tau^\dagger : \mathcal{H}_c \to \mathcal{H}_a$ is the adjoint of $\tau$. For a general mixed state input $\rho$ on system $a$, the output state up to normalization becomes $\tau\rho\tau^\dagger$, with the corresponding post-selection probability $\mathrm{Tr}[\langle\psi_2|_{ab} \, (\rho \otimes |\psi_1\rangle_{bc} \, \langle\psi_1|_{bc}) \, |\psi_2\rangle_{ab}] = \mathrm{Tr}[\tau\rho\tau^\dagger]$.

Denote the dimensions of $\mathcal{H}_a$, $\mathcal{H}_b$, and $\mathcal{H}_c$ by $d_a$, $d_b$, and $d_c$, respectively. We apply the Schmidt decomposition to the given states $|\psi_1\rangle_{bc}$, $|\psi_2\rangle_{ab}$, and the transition matrix $\tau$. There exist six orthonormal bases such that

$$|\psi_1\rangle_{bc} = \sum_{i=1}^{d_1} x_i \, |\beta_i\rangle_b \, |\gamma_i\rangle_c \,, \quad |\psi_2\rangle_{ab} = \sum_{i=1}^{d_2} y_i \, |\alpha_i\rangle_a \, |\tilde{\beta}_i\rangle_b \,, \tag{3}$$

$$\tau = \sum_{i=1}^{d} z_i \, |\tilde{\gamma}_i\rangle_c \, \langle\tilde{\alpha}_i|_a \,, \quad \tau^\dagger = \sum_{i=1}^{d} z_i \, |\tilde{\alpha}_i\rangle_a \, \langle\tilde{\gamma}_i|_c \,, \tag{4}$$

where we denote $d_1 = \min(d_b, d_c)$, $d_2 = \min(d_a, d_b)$ and $d = \min(d_a, d_b, d_c)$, and the coefficients $x_i$, $y_i$ and $z_i$ are real, non-negative numbers, unique up to re-ordering. $\tau$ is Hermitian and positive semi-definite if and only if $|\tilde{\gamma}_i\rangle = |\tilde{\alpha}_i\rangle$ for all the $i$ with $z_i > 0$. If there exists $|\tilde{\gamma}_i\rangle \neq \pm |\tilde{\alpha}_i\rangle$ for some $i$ whose $z_i > 0$, $\tau$ is non-Hermitian. In contrast, if there only exists $|\tilde{\gamma}_i\rangle = -|\tilde{\alpha}_i\rangle$ for some $i$ whose $z_i > 0$, then $\tau$ is Hermitian but not positive semi-definite.

The vector in these bases is unique up to reordering if its corresponding coefficient is nonzero. In general, the overlap matrix $\langle\tilde{\beta}_i|\beta_j\rangle \neq \delta_{ij}$ up to reordering, which implies that the bases $\{|\gamma_i\rangle\}$ and $\{|\tilde{\gamma}_i\rangle\}$ (and likewise for $\{|\alpha_i\rangle\}$) are distinct. The discussion for distillation of transition matrices in Sec. 3 will simplify in the diagonal case, where $|\beta_i\rangle_b = |\tilde{\beta}_i\rangle_b$. In this case, the transition matrix $\tau$ in (4) becomes

$$\tau \overset{\text{diag.}}{=} \sum_{i=1}^{d} z_i \, |\gamma_i\rangle_c \, \langle\alpha_i|_a \,, \quad z_i \overset{\text{diag.}}{=} x_i y_i \,. \tag{5}$$

Unlike in perfect quantum teleportation, where the transition matrix $\tau$ is proportional to the identity operator, general post-selection yields a nontrivial $\tau$, meaning only partial information of the input state is transferred from $a$ to $c$. In what follows, we introduce and compare several entropy measures of the transition matrix $\tau$ to quantify the amount of information transferred. In the diagonal case, the entropy measures of $\tau$ are directly related to the entanglement distillation of the states $|\psi_1\rangle_{bc}$ and $|\psi_2\rangle_{ab}$. Otherwise, their relation becomes more complicated due to the contribution of the non-identical matrix $\langle\tilde{\beta}_i|\beta_j\rangle$.

## 2.2 Entropy measures of transition matrix

In this section, we define the ABB entropy to describe the information transfer in the above post-selection process with $\tau$ and compare it to other measures of entropy.

### 2.2.1 ABB entropy

We aim to investigate how much information from the input state $\rho$ on system $a$ is transferred into the normalized output state $\rho_\tau = \tau\rho\tau^\dagger/\mathrm{Tr}[\tau\rho\tau^\dagger]$ via the post-selection process governed by the transition matrix $\tau$. The information transfer from $a$ to $c$ can be quantified by the minimal relative entropy between the input and output states, optimized over all unitary transformation $U$

$$\min_U S[U\rho_\tau U^\dagger || \rho] = \mathrm{Tr}[\rho_\tau \ln \rho_\tau] - \min_U \mathrm{Tr}[U\rho_\tau U^\dagger \ln \rho]. \tag{6}$$

Since unitary operations should not affect the amount of information transferred, we rule it out in the relative entropy by minimization. Clearly, this expression depends on both $\tau$ and $\rho$. To isolate the effect of $\tau$, we focus on the scenario in which the input state is the maximally mixed state (MMS) $\rho^{\mathrm{max}} = \mathbb{I}/d_a$, with $\mathbb{I}$ being the identity matrix in dimension $d_a$. In this case, the output state becomes

$$\rho_\tau^{\mathrm{max}} = \frac{\tau\rho^{\mathrm{max}}\tau^\dagger}{\mathrm{Tr}[\tau\rho^{\mathrm{max}}\tau^\dagger]} = \frac{\tau\tau^\dagger}{\mathrm{Tr}[\tau\tau^\dagger]} = \tilde{\tau}\tilde{\tau}^\dagger = \sum_{i=1}^{d} p_i \, |\tilde{\gamma}_i\rangle_c \, \langle\tilde{\gamma}_i|_c \,, \tag{7}$$

where we introduce a normalization for $\tau$, given by

$$\tilde{\tau} = \frac{\tau}{\sqrt{\mathrm{Tr}[\tau\tau^\dagger]}} = \sum_{i=1}^{d} \sqrt{p_i} \, |\tilde{\gamma}_i\rangle_c \, \langle\tilde{\alpha}_i|_a \,, \tag{8}$$

with $p_i = \frac{z_i^2}{\sum_j z_j^2}$ and $\sum_i p_i = 1$. This decomposition in the last step follows from the singular value decomposition of $\tau$ in (4). Now, the minimal relative entropy between $\rho_\tau^{\mathrm{max}}$ and $\rho^{\mathrm{max}}$ is

$$\min_U S[U\rho_\tau^{\mathrm{max}}U^\dagger || \rho^{\mathrm{max}}] = \mathrm{Tr}[\rho_\tau^{\mathrm{max}} \ln \rho_\tau^{\mathrm{max}}] + \ln d_a. \tag{9}$$

The first term is the negative von Neumann entropy of the output state $\rho_\tau^{\mathrm{max}}$, which is defined as

$$S_{\mathrm{von}}[\rho_\tau^{\mathrm{max}}] = -\mathrm{Tr}[\tilde{\tau}\tilde{\tau}^\dagger \ln(\tilde{\tau}\tilde{\tau}^\dagger)] = -\sum_{i=1}^{d} p_i \ln p_i = S_{\mathrm{ABB}}[\tau]. \tag{10}$$

The MMS $\rho^{\mathrm{max}}$ has the largest von Neumann entropy $\ln d_a$. After post-selection with transition matrix $\tau$, the information loss is reflected in the decrease of $S_{\mathrm{von}}[\rho_\tau^{\mathrm{max}}]$. A large (small) value of $S_{\mathrm{von}}[\rho_\tau^{\mathrm{max}}]$ indicates that most of the states in $\rho^{\mathrm{max}} = \frac{1}{d_a}\sum_{i=1}^{d_a} |\tilde{\alpha}_i\rangle_a \langle\tilde{\alpha}_i|_a$ survive (are eliminated) after post-selection, where we have used the basis $\{|\tilde{\alpha}_i\rangle\}$ coming from the decomposition of $\tau$ in (4). In the last equality of (10), we identify that this quantity coincides with the ABB entropy $S_{\mathrm{ABB}}[\tau]$, originally introduced in [70] to quantify the complexity of genome expression data. The name "ABB" is derived from the initials of the authors of [70], following the convention in [67].

Alternatively, based on the Choi–Jamiołkowski (CJ) isomorphism [77,78], we can map the transition matrix $\tau$ in (4) onto an entangled state $|\tau\rangle = \sum_i^d z_i \, |\tilde{\gamma}_i\rangle_c \, |\tilde{\alpha}_i\rangle_a$ on the joint system $ca$. Thus, the normalization of $\tau$ in (8) can be interpreted as the normalization of the state $|\tau\rangle$, allowing the ABB entropy in (10) to be interpreted as the entanglement entropy between system $c$ and $a$ on the state $|\tau\rangle$ after normalization. Finally, the corresponding Rényi ABB entropy of $\tau$ is given by

$$S_{\mathrm{ABB}}^{(n)}[\tau] = \frac{1}{1-n} \ln \frac{\mathrm{Tr}\big[\big(\tau^\dagger \tau\big)^n\big]}{\mathrm{Tr}[\tau^\dagger \tau]^n} = \frac{1}{1-n} \ln \mathrm{Tr}\left(\tilde{\tau}^\dagger \tilde{\tau}\right)^n. \tag{11}$$

Obviously, the ABB entropy and its Rényi version are invariant under the bi-unitary transformations $\tau \to U\tau V$ with arbitrary unitary matrices $U$ and $V$.

We now review three other notions of entropy defined for the transition matrix $\tau$ below, namely the (modified) pseudo entropy [16, 56] and the SVD entropy [67].

### 2.2.2 (Modified) pseudo entropy

Several attempts have been made to generalize the von Neumann entropy for density matrices to describe the "entropy" of non-Hermitian matrices [79] when systems $a$ and $c$ could be identical. In our paper, when the Hilbert spaces of $\mathcal{H}_a$ and $\mathcal{H}_c$ have the same dimension, given a transition matrix $\tau$ in (1), we define a linear map from $\mathcal{H}_a$ to $\mathcal{H}_a$ by teleporting the outcome state $\tau \, |\phi\rangle_a$ from system $c$ back to system $a$ via the perfect quantum teleportation protocol [1, 2, 80] that teleports the basis state $|\gamma_i\rangle_c$ to $|\gamma_i\rangle_a$. Mathematically, this is equivalent to identifying the system $a$ and system $c$ from the beginning. One considers two states $|\psi_1\rangle_{ab}$ and $|\psi_2\rangle_{ab}$ and defines the linear map as

$$\tau = \mathrm{Tr}_b \, |\psi_1\rangle_{ab} \, \langle\psi_2|_{ab} = \sum_{i=1}^d z_i \, |\tilde{\gamma}_i\rangle_a \, \langle\tilde{\alpha}_i|_a : \mathcal{H}_a \to \mathcal{H}_a \,, \tag{12}$$

where we have abused the notation of $\tau$. Actually, Eq. (12) is the original definition of a transition matrix as given in [16].

When $|\psi_1\rangle_{ab} = |\psi_2\rangle_{ab}$, the transition matrix $\tau = \mathrm{Tr}_b \, |\psi_2\rangle \, \langle\psi_2|$ is Hermitian, positive semi-definite, and has a unit trace. It is mathematically identical to a reduced density matrix on the system $a$. However, it is crucial to distinguish the physical roles: a transition matrix describes an operation on an unknown quantum state, while a density matrix represents a quantum state itself. Although one could, as in [16], disregard the interpretation of $\tau$ as a Hermitian transition matrix rather than a density matrix and directly apply the von Neumann entropy formula, $-\mathrm{Tr}[\tau \ln \tau]$, the physical significance of such an entropy remains unclear. This is because the interpretation of von Neumann entropy for a density matrix cannot be directly extended to a transition matrix.

When $|\psi_1\rangle_{ab} \neq |\psi_2\rangle_{ab}$, $\tau$ generally becomes non-Hermitian and has non-unity or even complex trace. It therefore cannot be regarded as a density matrix, which manifests the fundamental distinction between transition matrices and density matrices. Nevertheless, following [16] and assuming $\mathrm{Tr}\,\tau \neq 0$, one can define a normalized transition matrix

$$\hat{\tau} = \frac{\tau}{\mathrm{Tr}\,\tau} = \sum_{i=1}^d \lambda_i \, |r_i\rangle \, \langle l_i| \,, \tag{13}$$

where $\hat{\tau}$ is generally non-Hermitian and thus admits a bi-orthogonal decomposition [49]. The spectrum $\{\lambda_i\}$ is generally complex, and the right and left eigenbasis $\{|r_i\rangle\}$, $\{|l_i\rangle\}$

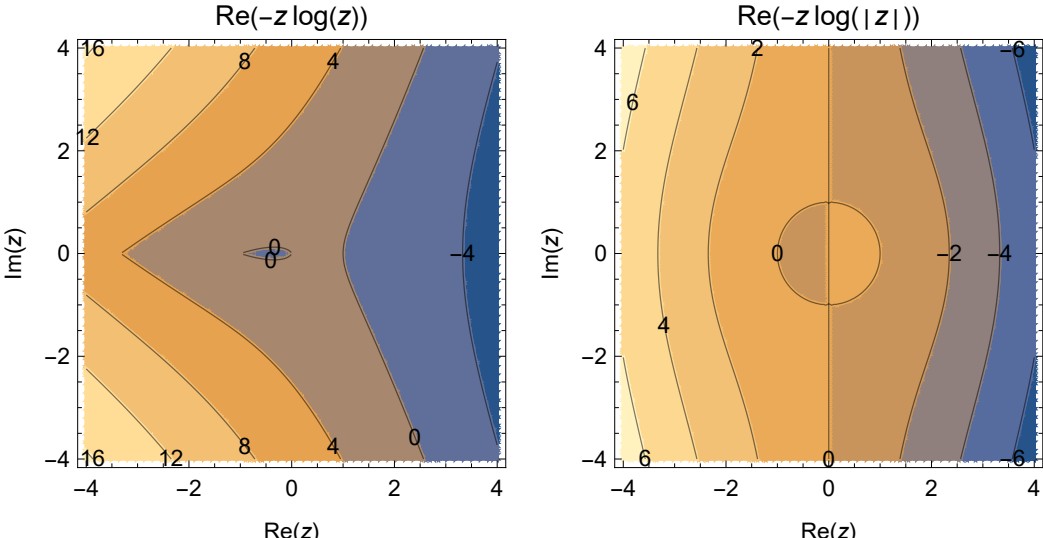

Figure 2: The real parts of two entropy formulas $\mathrm{Re}[-z \ln z]$ and $\mathrm{Re}[-z \ln |z|]$ on the complex plane.

satisfy the bi-orthogonality condition $\langle l_i | r_i \rangle = \delta_{ij}$, $\forall i, j$. Plugging the spectrum of $\hat{\tau}$ into the formulas of the von Neumann and Rényi entropies leads to the definition of the pseudo entropy,

$$S_{\mathrm{P}}[\tau] = -\mathrm{Tr}\,\hat{\tau}\ln\hat{\tau} = -\sum_{i=1}^{d} \lambda_i \ln \lambda_i, \tag{14}$$

and the Rényi pseudo entropy,

$$S_{\mathrm{P}}^{(n)}[\tau] = \frac{1}{1-n} \ln \mathrm{Tr}\,\hat{\tau}^n = \frac{1}{1-n} \ln \sum_{i=1}^{d} \lambda_i^n. \tag{15}$$

To compute the (modified) pseudo entropy, the Schmidt bases $\{|\tilde{\gamma}_i\rangle\}$, $\{|\tilde{\alpha}_j\rangle\}$ in (4) are not useful in general, since they are not necessarily bi-orthogonal. For Hermitian $\hat{\tau}$, the right and left eigenbasis are identical. From the Schmidt decomposition in (4), we can observe that when $\hat{\tau}$ is Hermitian, such as $|\tilde{\gamma}_i\rangle = e^{i\theta} |\tilde{\alpha}_i\rangle$ with a global phase $\theta$ for all $i$ with $z_i > 0$, the pseudo entropy is real and positive. However, in general, $\hat{\tau}$ can either remain Hermitian with negative eigenvalues included or become non-Hermitian. Except for some pseudo-Hermitian cases, the spectrum of a non-Hermitian $\tau$ must include complex values. In this case, the normalization factor $\mathrm{Tr}\,\tau$ can become negative, complex, nearly zero or even vanish. This leads to serious issues, such as the emergence of negative and complex-valued entropies. Furthermore, when $\mathrm{Tr}\,\tau$ is close to zero or vanishes, the absolute value of pseudo entropy can be significantly enhanced, or diverge. We will show that it is a common phenomenon near the exceptional points in $PT$-symmetric non-Hermitian systems in Sec. 6.

For the non-positive spectrum of $\hat{\tau}$, $\ln\hat{\tau}$ in (14) is a multi-valued function, resulting in multi-valued pseudo entropies. In critical non-Hermitian systems governed by non-unitary CFTs with negative central charges, the transition matrix $\tau$ made of bi-orthogonal basis leads to negative-valued pseudo entropy, which exactly accommodates with the negative central charge. In [50], a proper branch of $\ln\hat{\tau}$ was artificially chosen to obtain the desired

negative central charge in a non-Hermitian SSH model at a critical point. Complementary to this approach, a modified version of the entropy formula was proposed in [56] as follows [1],

$$S_{\mathrm{MP}}[\tau] = -\mathrm{Tr}\,\hat{\tau} \ln |\hat{\tau}| = -\sum_{i=1}^{d} \lambda_i \ln |\lambda_i|, \qquad (16)$$

where the $|\hat{\tau}|$ is defined by taking the modulus of the spectrum of $\hat{\tau}$ as $|\hat{\tau}| = \sum_{i=1}^{d} |\lambda_i| \times |r_i\rangle \langle l_i|$. The corresponding Rényi modified pseudo entropy is given by

$$S_{\mathrm{MP}}^{(n)}[\tau] = \frac{1}{1-n} \ln \mathrm{Tr}\,\left(\hat{\tau}\,|\hat{\tau}|^{n-1}\right) = \frac{1}{1-n} \ln \left(\sum_{i=1}^{d} \lambda_i\,|\lambda_i|^{n-1}\right). \qquad (17)$$

The (modified) pseudo entropy and its Rényi version are invariant only under unitary similarity transformations $\tau \to U\tau U^\dagger$.

This modified definition successfully captures the expected negative central charges in several critical non-Hermitian models, including the two-legged SSH model and the $q$-deformed XXZ model [56]. While $\ln |\hat{\tau}|$ in (16) is always real, the modified pseudo entropy can still become complex when the complex spectrum lacks specific symmetry properties. However, if the spectrum of $\hat{\tau}$ is real or forms complex conjugate pairs, then the entropy remains real, though not necessarily positive. Moreover, the issue of tiny and vanishing $\mathrm{Tr}\,\tau$ remains unresolved in the modified pseudo entropy, meaning that it is still not bounded by the logarithm of the Hilbert space dimension.

Finally, when dealing with random matrix models, as in Sec. 4 and 5, the ensemble-averaged (modified) pseudo entropy generally reduces to the contribution from the real part, as the imaginary part typically averages out to zero. Given this, we present the value distributions of two entropy formulas $\mathrm{Re}[-z \ln z]$ and $\mathrm{Re}[-z \ln |z|]$ on the complex plane in Fig. 2.

### 2.2.3 SVD entropy

As mentioned earlier, the pseudo entropy generally exhibits nonphysical behavior, such as complex values or divergencies, due to the general non-positivity or complex nature of the spectrum of $\tau$. To address these issues, a natural alternative to pseudo entropy is the SVD entropy [67], which is based on the singular value decomposition (SVD) of $\tau$. This approach constructs a novel normalized transition matrix via SVD-normalization as

$$\bar{\tau} = \frac{\sqrt{\tau^\dagger \tau}}{\mathrm{Tr}\sqrt{\tau^\dagger \tau}} = \sum_{i=1}^{d} q_i\,|\tilde{\alpha}_i\rangle_a\,\langle\tilde{\alpha}_i|_a\,, \qquad (18)$$

where the normalized sequence is defined as $q_i = \frac{z_i}{\sum_i z_i}$, with $z_i$ being singular values of $\tau$ in (4). Notably, $\bar{\tau}$ in (18) is formally a density matrix defined on the system $a$, with the transition information from $a$ to $c$ already being encoded in the sequence $\{q_i\}$. Following [67], we give the definition of SVD entropy of $\tau$ as

$$S_{\mathrm{SVD}}[\tau] = -\mathrm{Tr}\,\bar{\tau} \ln \bar{\tau}. \qquad (19)$$

---

[1] In general, here the $|\hat{\tau}|$ is different from $\sqrt{\hat{\tau}^\dagger \hat{\tau}}$ nor $\sqrt{\hat{\tau}\hat{\tau}^\dagger}$, except when $\hat{\tau}$ is Hermitian. So the modified pseudo entropy could be considered as a generalization of the FP entropy defined in [64].

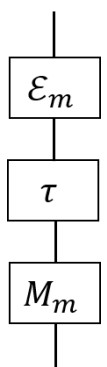

Figure 3: Measurement and non-unitary operation

The SVD entropy is real, non-negative, and strictly bounded by the Hilbert space dimension. Nevertheless, no concrete bound exists between the SVD entropy of $\tau$ and the corresponding entanglement entropies of two states $|\psi_1\rangle_{bc}$ and $|\psi_2\rangle_{ab}$. In analogy with entanglement entropy, a Rényi version of the SVD entropy for $\tau$ is defined by

$$S_{\text{SVD}}^{(n)}[\tau] = \frac{1}{1-n} \ln \frac{\text{Tr}[\sqrt{\tau^\dagger \tau}^n]}{\text{Tr}[\sqrt{\tau^\dagger \tau}]^n} = \frac{1}{1-n} \ln \text{Tr}\, \bar{\tau}^n. \tag{20}$$

The SVD entropy in integrable CFTs, holographic CFTs and Chern-Simons theory was explored in [67], but due to existence of the square root in $\bar{\tau}$, it is difficult to deal with the field-theoretic analysis in the path-integral formalism. To address this issue, an extension of the replica trick by introducing the second replica index $m$ is used,

$$S_{\text{SVD}}^{(n,m)}[\tau] = \frac{1}{1-n} \ln \frac{\text{Tr}[(\tau^\dagger \tau)^{\frac{mn}{2}}]}{\text{Tr}[(\tau^\dagger \tau)^{\frac{m}{2}}]^n}, \tag{21}$$

whose value at odd $m$ should be obtained by the analytical continuation of $m$ from even integers due to the fractional powers [67]. The SVD entropy is obtained in the limit

$$S_{\text{SVD}} = \lim_{n \to 1} \lim_{m \to 1} S_{\text{SVD}}^{(n,m)}. \tag{22}$$

The SVD entropy and its Rényi version are invariant under bi-unitary transformation $\tau \to U\tau V$.

## 2.3 Entropy monotone

For quantum states, the entanglement transformation has been an important topic in quantum information theory. The connection between entanglement transformation and the linear algebraic theory of "majorization" was made in [71], which states that an entangled state $|\psi\rangle$ can be transformed into another one $|\phi\rangle$ by local operations and classical communication (LOCC) if and only if the spectra of their respective reduced density matrices satisfy the majorization relation $\lambda_\psi \prec \lambda_\phi$ [2]. For the $d$-dimensional vectors $\lambda_\psi = (x_1, \cdots, x_d)$ and $\lambda_\phi = (y_1, \cdots, y_d)$, each arranged in descending order, the

---

[2]Beside the density matrix majorization, one can also discuss the relation between two states in terms of Wigner majorization in the phase space [81–83].

statement that $\lambda_\psi$ is majorized by $\lambda_\phi$ or $\lambda_\phi$ majorizes $\lambda_\psi$, means that [84]

$$\sum_{i=1}^{k} x_i \leq \sum_{i=1}^{k} y_i, \quad \text{for} \quad k = 1, \cdots, d, \tag{23}$$

with equality holding when $k = d$. Since the von Neumann entropy is a Schur-concave function $S : R^d \to R$, the majorization relation $\lambda_\psi \prec \lambda_\phi$ implies $S[\lambda_\psi] \geq S[\lambda_\phi]$. In other words, the entanglement entropy of a pure state between its two subsystems will always decrease under LOCC. Other entanglement monotonicity under LOCC and also the case of multipartite states were explored in [85–87]. More recently, a LOCC theory for bipartite systems was developed by commuting von Neumann algebras [88], in which the central result states that the LOCC ordering of bipartite pure states is equivalent to the majorization of their restrictions.

As mentioned in Sec. 2.2.1, since the state-normalized transition matrix $\tilde{\tau}$ corresponds to an entangled state, we explore the analogous relation between local operations, majorization, and entropy transformation for transition matrices in this section.

We consider the transformation from transition matrix $\tilde{\tau}$ to another transition matrix $\tilde{t}$ by first applying a generalized measurement $M_m$ and then applying an operation $\mathcal{E}_m$, as depicted in Fig. 3, where the generalized measurement $M_m$ transformation satisfies $\sum_m M_m M_m^\dagger = I$ and the operation $\mathcal{E}_m$ is a (possibly non-unitary) trace-preserving map depending on the measurement outcome $m$. The resulting relation between $\tilde{\tau}$ and $\tilde{t}$ is thus given by

$$\tilde{t} \otimes \tilde{t}^\dagger = \sum_m \mathcal{E}_m(\tilde{\tau} M_m \otimes M_m^\dagger \tilde{\tau}^\dagger), \tag{24}$$

where action of $\tilde{t} \otimes \tilde{t}^\dagger$ is to transform any density matrix $\rho$ into $\tilde{t} \rho \tilde{t}^\dagger$. Analogous to the requirement of the LOCC resulting in pure state [71], here due to the factorized structure on the left-hand side of (24), each term on the right-hand side is proportional to $\tilde{t} \otimes \tilde{t}^\dagger$, i.e., $P_m \tilde{t} \otimes \tilde{t}^\dagger = \mathcal{E}_m(\tilde{\tau} M_m \otimes M_m^\dagger \tilde{\tau}^\dagger)$ with $P_m$ being an undetermined probability constrained by the normalized condition $\sum_m P_m = 1$. Taking the trace over system $c$, or rather, performing the contraction over the second index of $\tilde{t}$, and using the trace-preserving property of $\mathcal{E}_m$, we obtain

$$P_m \text{Tr}_c[\tilde{t} \otimes \tilde{t}^\dagger] = \text{Tr}_c[\mathcal{E}_m(\tilde{\tau} M_m \otimes M_m^\dagger \tilde{\tau}^\dagger)] \Rightarrow P_m \tilde{t}^\dagger \tilde{t} = M_m^\dagger \tilde{\tau}^\dagger \tilde{\tau} M_m. \tag{25}$$

Applying the polar decomposition [89]

$$M_m^\dagger \sqrt{\tilde{\tau}^\dagger \tilde{\tau}} = \sqrt{M_m^\dagger \tilde{\tau}^\dagger \tilde{\tau} M_m} U_m = \sqrt{P_m} \sqrt{\tilde{t}^\dagger \tilde{t}} U_m, \tag{26}$$

we find

$$\tilde{\tau}^\dagger \tilde{\tau} = \sum_m \sqrt{\tilde{\tau}^\dagger \tilde{\tau}} M_m M_m^\dagger \sqrt{\tilde{\tau}^\dagger \tilde{\tau}} = \sum_m P_m U_m^\dagger \tilde{t}^\dagger \tilde{t} U_m. \tag{27}$$

The eigen spectra of $\tilde{\tau}^\dagger \tilde{\tau}$ and $\tilde{t}^\dagger \tilde{t}$ are denoted by sequences $p[\tilde{\tau}]$ and $p[\tilde{t}]$, respectively. Sorting the elements of both sequences in descending order with $p_i[\tilde{\tau}] \geq p_{i+1}[\tilde{\tau}]$ and $p_i[\tilde{t}] \geq p_{i+1}[\tilde{t}]$, the relation (27) implies the sequences $p[\tilde{\tau}]$ is majorized by $p[\tilde{t}]$, i.e.,

$$p[\tilde{\tau}] \prec p[\tilde{t}]. \tag{28}$$

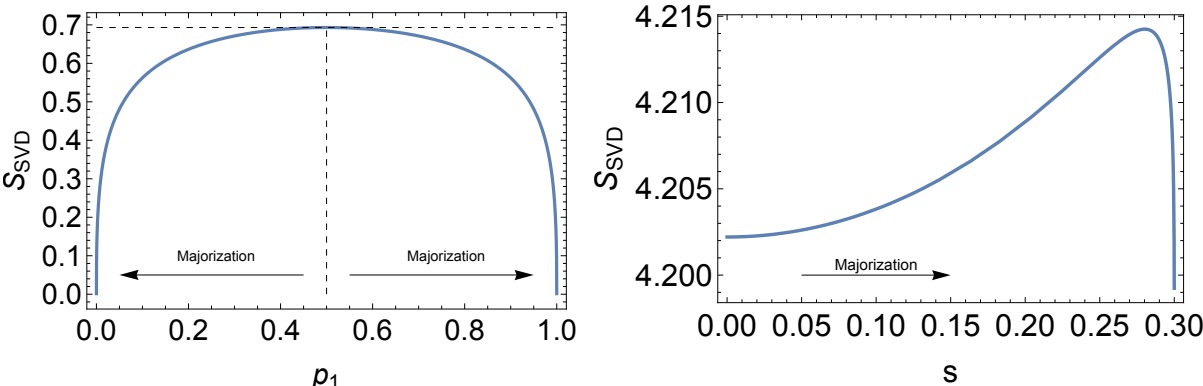

Figure 4: Left: SVD entropy in a two-dimensional Hilbert space. Two majorization paths are shown: one from $p_1 = 0.5$ to 0, and another from $p_1 = 0.5$ to 1 with $p_2 = 1 - p_1$. Along both paths, the SVD entropy decreases monotonically. Right: SVD entropy in an 80-dimensional Hilbert space. Along the majorization path parameterized by $p_1 = 3/10 + s$, $p_2 = 3/10 - s$, and $p_3 = p_4 = \cdots = p_{80} = 1/195$, the SVD entropy increases over a broad range of $s$.

This is called Nielsen's theorem, whose rigorous proof can be found in [90]. Since the von Neumann entropy $S_{\text{von}}[p] = -\sum_i p_i \log p_i$ is Schur-concave with respect to the spectrum of $p$ [89], the majorization in (28) ensures that the ABB entropy of $\tau$ always decreases under the measurement and non-unitary operation in (24), i.e.,

$$S_{\text{ABB}}[\tau] \geq S_{\text{ABB}}[t]. \tag{29}$$

Next, we examine whether a similar monotonicity holds for the SVD entropy. From (18) and (8), we can easily find that the spectrum of $\bar{\tau}$ can be expressed as $q_i = \frac{\sqrt{p_i}}{\sum_j \sqrt{p_j}}$, which does not necessarily imply $q[\tilde{\tau}] \prec q[\tilde{t}]$, in spite of (28). Now, we can consider a majorization path in the spectrum space $\{p_i\}$, parameterized by the increasing variable $s$, along which $p[s_1] \succ p[s_2]$ for $s_1 > s_2$. Then we can study how the SVD entropy behaves along the majorization path. The SVD entropy can be expanded as

$$S_{\text{SVD}}[\tau] = -\sum_i q_i \log q_i = \frac{1}{\sum_j \sqrt{p_j}} \left[ -\sum_i \sqrt{p_i} \log \sqrt{p_i} + \sum_i \sqrt{p_i} \log \left( \sum_j \sqrt{p_j} \right) \right]. \tag{30}$$

We numerically verify that the SVD entropy is a Schur-concave function of $p$ in low-dimensional Hilbert spaces. For instance, in a two-dimensional Hilbert space, as shown in the left panel of Fig. 4, it decreases monotonically along majorization paths. Similarly, the Schur concavity also holds in three-dimensional space. However, this does not necessarily hold in higher-dimensional Hilbert spaces, such as the majorization path in 80-dimensional Hilbert space shown in the right panel of Fig. 4, the SVD entropy increases along the majorization path in a region, violating the Schur concavity. Moreover, this observation is also confirmed by the Schur-concavity criterion, which states that for any $i$, $j$, $\Delta = (p_i - p_j) \left( \frac{\partial S}{\partial p_i} - \frac{\partial S}{\partial p_j} \right) \leq 0$ [91]. Thus, we conclude that the SVD entropy is not universally Schur-concave for the probability distribution $\{p_i\}$ in arbitrary dimensions.

When the (modified) pseudo entropy coincides with the SVD entropy for a Hermitian $\tau$, the (modified) pseudo entropy also fails to be Schur-concave, as illustrated in Fig. 4. If the (modified) pseudo entropy takes a complex value, the notion of Schur concavity is no

longer meaningful. Later, in Sec. 6.1, we will see that in the $PT$-symmetric region, the (modified) pseudo entropy is real but differs from the SVD entropy. In contrast, since $\tilde{\tau}\tilde{\tau}^\dagger = \bar{\tau} = \mathbb{I}/2$, the ABB entropy, which happens to coincide with the SVD entropy, remains constant. Meanwhile, the pseudo entropy increases as $\mu$ decreases, while the modified pseudo entropy decreases with decreasing $\mu$, indicating that neither of them is Schur-concave.

In summary, we have demonstrated that transition matrix $\tilde{\tau}$ can be transformed into another transition matrix $\tilde{t}$ via generalized measurement and non-unitary operation if and only if the spectra of $\tilde{\tau}^\dagger\tilde{\tau}$ and $\tilde{t}^\dagger\tilde{t}$ satisfy the majorization relation $p[\tilde{\tau}] \prec p[\tilde{t}]$, which implies $S_{\mathrm{ABB}}[\tau] \geq S_{\mathrm{ABB}}[t]$. However, this monotonicity does not generally hold for the SVD and pseudo entropies. Therefore, in the context of entropy transformation under LOCC-like operation, the ABB entropy of $\tau$ offers a more physically reasonable measure.

# 3 Probabilistic interpretation

In this section, we will explore the probabilistic interpretation of the ABB entropy from the perspective of distillation. To do that, we first review the entanglement distillation of $|\psi_1\rangle_{bc}$ and $|\psi_2\rangle_{ab}$ for arbitrary dimensions by generalizing the formalism of [69], which describes the process of transforming the $m$-copy entangled states into some amount of EPR pairs through LOCC [69]. The entanglement entropy $S$ in base 2 measures the large-$m$ asymptotic number of distilled EPR pairs from each copy of the system.

Following this perspective, we extend the entanglement distillation from pure states to the transition matrix $\tilde{\tau}$ in (8). Our analysis reveals that only the distillation of the ABB entropy has a well-defined probabilistic interpretation from the construction of the transition matrix.

Finally, we revisit the probabilistic interpretation of pseudo entropy [16] and SVD entropy [67], based on the construction of transition matrices $\hat{\tau}$ in (13) and $\bar{\tau}$ in (18). We are unable to find a clear probabilistic interpretation of the pseudo entropy and SVD entropy based on the construction.

## 3.1 Entanglement distillation of quantum state

We consider the states $|\psi_1\rangle_{bc}$ and $|\psi_2\rangle_{ab}$ in (3). We focus on entanglement distillation from $m$ copies of $|\psi_1\rangle_{bc}$ first, and similarly for $|\psi_2\rangle_{ab}$. Given the Schmidt decomposition of $|\psi_1\rangle_{bc}$ in terms of the $d_1$ pairs of basis vectors $\{|\beta_i\rangle_b |\gamma_i\rangle_c\}_{i=1}^{d_1}$ in (3), we define a $d_1$-dimensional subspace spanned by these pairs of basis vectors, namely $\mathcal{H}_1 = \mathrm{span}\,\{|\beta_i\rangle_b |\gamma_i\rangle_c\}_{i=1}^{d_1}$. The $m$-copy state $|\psi_1\rangle_{bc}^{\otimes m}$ belongs to the $m$-copy subspace $\mathcal{H}_1^{\otimes m}$. We further decompose this subspace into the direct sum of $\binom{m+d_1-1}{d_1-1}$ orthogonal subspaces,

$$\mathcal{H}_1^{\otimes m} = \oplus_{\mathbf{k}}\mathcal{H}_{1\mathbf{k}}, \quad \text{with } \mathbf{k} = (k_1, k_2, \cdots, k_{d_1}),\ \sum_{i=1}^{d_1} k_i = m\,, \tag{31}$$

based on the multinomial expansion $d_1^m = \sum_{\mathbf{k}} d_{1\mathbf{k}}$, where the multinomial coefficient $d_{1\mathbf{k}} = \binom{m}{\mathbf{k}} = \frac{m!}{k_1!k_2!\cdots k_{d_1}!}$ is the dimension of the subspace $\mathcal{H}_{1\mathbf{k}}$. The number of different configurations of $\mathbf{k}$ is $\binom{m+d_1-1}{d_1-1}$. Accordingly, the state $|\psi_1\rangle_{bc}^{\otimes m}$ can be expressed as a

superposition of $\binom{m+d_1-1}{d_1-1}$ MESs $|BC_{\mathbf{k}}\rangle$ over each subspace $\mathcal{H}_{1\mathbf{k}}$,

$$|\psi_1\rangle_{bc}^{\otimes m} = \sum_{\mathbf{k}} \sqrt{P_{1\mathbf{k}}} |BC_{\mathbf{k}}\rangle \,, \ |BC_{\mathbf{k}}\rangle = \frac{1}{\sqrt{d_{1\mathbf{k}}}} \sum_{\mu=1}^{d_{1\mathbf{k}}} |B_{\mu}^{\mathbf{k}}\rangle |C_{\mu}^{\mathbf{k}}\rangle \,, \ P_{1\mathbf{k}} = d_{1\mathbf{k}} \prod_{i=1}^{d_1} x_i^{2k_i}, \qquad (32)$$

where the state $|B_{\mu}^{\mathbf{k}}\rangle$ is a rank-$m$ tensor product of $k_1$ copies of $|\beta_1\rangle$, $k_2$ copies of $|\beta_2\rangle$, $\cdots$, $k_{d_1}$ copies of $|\beta_{d_1}\rangle$ for a fixed configuration $\mathbf{k}$, and the index $\mu$ labels different orders of these bases in the tensor product. Similarly, the state $|C_{\mu}^{\mathbf{k}}\rangle$ is a tensor product of $\{|\gamma_i\rangle\}$. An incomplete von Neumann measurement projecting onto the subspace $\mathcal{H}_{1\mathbf{k}}$ causes $|\psi_1\rangle_{bc}^{\otimes m}$ to collapse into $|BC_{\mathbf{k}}\rangle$ with probability $P_{1\mathbf{k}}$, as given in (32). The entanglement of a MES, e.g., $|BC_{\mathbf{k}}\rangle$, which has $d_{1\mathbf{k}}$ equally weighted terms in its Schmidt decomposition, can be equivalently regarded as that of $\log_2 d_{1\mathbf{k}}$ EPR pairs.

In the limit $m \to \infty$, the incomplete von Neumann measurement causes the initial state to collapse into the dominant term with highest probability among all $\binom{m+d_1-1}{d_1-1}$ MESs. We take the logarithm of $P_{1\mathbf{k}}$ and apply Stirling's approximation in the large-$m$ limit, thus obtaining

$$\ln P_{1\mathbf{k}} = \ln \left( \frac{m!}{k_1! \, k_2! \, \cdots \, k_{d_1}!} \prod_{i=1}^{d_1} x_i^{2k_i} \right) \approx m \ln m - \sum_{i=1}^{d_1} \ln \frac{k_i}{x_i^2}. \qquad (33)$$

To determine $\mathbf{k}^*$ for the most probable distribution $P_{1\mathbf{k}^*}$, we introduce a Lagrange multiplier $\mu$ to enforce the constraint $\sum_{i=1}^{d_1} k_i = m$ and define the function

$$\mathcal{F} = \ln P_{1\mathbf{k}} - \mu \left( \sum_{i=1}^{d_1} k_i - m \right). \qquad (34)$$

Taking the derivative with respect to any $k_i$ gives

$$\frac{\partial \mathcal{F}}{\partial k_i} = -\ln \frac{k_i}{x_i^2} + \mu - 1 = 0 \implies k_i = x_i^2 \, e^{\mu-1}. \qquad (35)$$

Using the constraint on $\mathbf{k}$ again yields $e^{\mu-1} = m$. Thus, the probability $P_{1\mathbf{k}}$ sharply peaks around $\mathbf{k}^* = \left( mx_1^2, \cdots, mx_{d_1}^2 \right)$. Consequently, the MESs $|BC_{\mathbf{k}^*}\rangle$ in dimension $d_{1\mathbf{k}^*}$ are distilled from $|\psi_1\rangle_{bc}^{\otimes m}$. The dimension of the MES offers an interpretation of the entanglement entropy via

$$\lim_{m \to \infty} \frac{\ln d_{1\mathbf{k}^*}}{m} = -\sum_{i=1}^{d_1} x_i^2 \ln x_i^2 = S_1, \qquad (36)$$

Namely, the entanglement entropy of a quantum state equals the logarithm of the dimension of the MES distilled from its infinite copies, normalized by the number of copies.

In the same approach, $|\psi_2\rangle_{ab}^{\otimes m}$ has the similar expansion

$$|\psi_2\rangle_{ab}^{\otimes m} = \sum_{\mathbf{l}} \sqrt{P_{2\mathbf{l}}} |AB_{\mathbf{l}}\rangle \,, \ |AB_{\mathbf{l}}\rangle = \frac{1}{\sqrt{d_{2\mathbf{l}}}} \sum_{\mu=1}^{d_{2\mathbf{l}}} |A_{\mu}^{\mathbf{l}}\rangle |B_{\mu}^{\mathbf{l}}\rangle \,, \ P_{2\mathbf{l}} = d_{2\mathbf{l}} \prod_{i=1}^{d_2} y_i^{2l_i}, \qquad (37)$$

where $\mathbf{l} = (l_1, l_2, \cdots, l_{d_2})$ satisfies the constraint $\sum_{i=1}^{d_2} l_i = m$, and the state $|A_{\mu}^{\mathbf{l}}\rangle$ is a tensor product of $\{|\alpha_i\rangle\}$. Likewise, $P_{2\mathbf{l}}$ peaks around $\mathbf{l}^* = \left( my_1^2, \cdots, my_{d_2}^2 \right)$, and the $d_{2\mathbf{l}^*}$-dimensional MESs $|AB_{\mathbf{l}^*}\rangle$ are distilled from $|\psi_2\rangle_{ab}^{\otimes m}$. The corresponding relation between

the entanglement entropy of $|\psi_2\rangle_{ab}$ and $|AB_{\mathbf{l}^*}\rangle$ is

$$\lim_{m\to\infty} \frac{\ln d_{2\mathbf{l}^*}}{m} = -\sum_{i=1}^{d_2} y_i^2 \ln y_i^2 = S_2. \tag{38}$$

Next, we extend this notion of distillation from pure states to transition matrices.

## 3.2 Distillation of transition matrices

In this subsection, we analyze the probabilistic interpretation of ABB entropy, (modified) pseudo entropy and SVD entropy from the distillation. Our goal is to demonstrate that the ABB entropy provides a more physically meaningful measure compared to the pseudo entropy and SVD entropy.

### 3.2.1 ABB entropy

Now we analyze the probabilistic interpretation of the ABB entropy based on the following two distillation methods.

In Sec. 2.2.1, the ABB entropy of $\tau$ is identified with the entanglement entropy of the normalized $|\tilde{\tau}\rangle$ between $a$ and $c$. Following Sec. 3.1, the entanglement distillation of $|\tilde{\tau}\rangle$ can offer an interpretation of the ABB entropy. Via the CJ isomorphism, the $m$-copy transition matrix $\tilde{\tau}^{\otimes m}$ corresponds to the $m$-copy state $|\tilde{\tau}\rangle^{\otimes m}$, both of which have the expansion based on (8),

$$\tilde{\tau}^{\otimes m} = \sum_{\mathbf{k}} \sqrt{P_{\mathbf{k}}}\, \tilde{T}_{\mathbf{k}}, \quad \tilde{T}_{\mathbf{k}} = \frac{1}{\sqrt{d_{\mathbf{k}}}} \sum_{\mu=1}^{d_{\mathbf{k}}} |\tilde{C}_\mu^{\mathbf{k}}\rangle\langle\tilde{A}_\mu^{\mathbf{k}}|, \tag{39}$$

$$|\tilde{\tau}\rangle^{\otimes m} = \sum_{\mathbf{k}} \sqrt{P_{\mathbf{k}}}\, |\tilde{C}\tilde{A}_{\mathbf{k}}\rangle, \quad |\tilde{C}\tilde{A}_{\mathbf{k}}\rangle = \frac{1}{\sqrt{d_{\mathbf{k}}}} \sum_{\mu=1}^{d_{\mathbf{k}}} |\tilde{C}_\mu^{\mathbf{k}}\rangle|\tilde{A}_\mu^{\mathbf{k}}\rangle, \tag{40}$$

where $\sum_{\mathbf{k}} d_{\mathbf{k}} = d^m$, and similarly, the state $|\tilde{C}_\mu^{\mathbf{k}}\rangle$ is a tensor product of $\{|\tilde{\gamma}_i\rangle\}$ and the state $|\tilde{A}_\mu^{\mathbf{k}}\rangle$ is a tensor product of $\{|\tilde{\alpha}_i\rangle\}$. The sub transition matrix $\tilde{T}_{\mathbf{k}}$ defines a rank-$d_{\mathbf{k}}$ isometric map from $a$ to $c$. Based on (8), (32) and (37), the probability $P_{\mathbf{k}}$ of distilling MES $|\tilde{C}\tilde{A}_{\mathbf{k}}\rangle$ from $|\tilde{\tau}\rangle^{\otimes m}$ can be written as

$$P_{\mathbf{k}} = d_{\mathbf{k}} \prod_{i=1}^{d} p_i^{2k_i} \stackrel{\text{diag.}}{=} \frac{P_{1\mathbf{k}} P_{2\mathbf{k}}/d_{\mathbf{k}}}{\sum_{\mathbf{j}} P_{1\mathbf{j}} P_{2\mathbf{j}}/d_{\mathbf{j}}}. \tag{41}$$

We observe that in the diagonal case, $P_{\mathbf{k}}$ can be expressed as the joint probability $P_{1\mathbf{k}}P_{2\mathbf{k}}$ of simultaneously distilling $d_{\mathbf{k}}$-dimensional MESs $|BC_{\mathbf{k}}\rangle$ from $|\psi_1\rangle_{bc}^{\otimes m}$ and $|AB_{\mathbf{k}}\rangle$ from $|\psi_2\rangle_{ab}^{\otimes m}$ with a suppression factor $1/d_{\mathbf{k}}$, where the factor accounts for the probability arising from the projection onto the subspace spanned by $\{|B_\mu^{\mathbf{k}}\rangle\}_{\mu=1}^{d_{\mathbf{k}}}$ during post-selection from $|BC_{\mathbf{k}}\rangle$ to $|AB_{\mathbf{k}}\rangle$. According to the same entanglement distillation process as described in Sec. 3.1, $|\tilde{\tau}\rangle^{\otimes m}$ concentrates on the state $|CA_{\mathbf{k}^*}\rangle$ with the highest probability $P_{\mathbf{k}^*}$ with $\mathbf{k}^* = (mp_1^2, \cdots, mp_d^2)$ in the large-$m$ limit. Correspondingly, the transition matrix $\tilde{\tau}$ also concentrates on the sub transition matrix $\tilde{T}_{\mathbf{k}^*}$ with probability $P_{\mathbf{k}^*}$. Thus, based on the same observation,

$$\lim_{m\to\infty} \frac{\ln d_{\mathbf{k}^*}}{m} = -\sum_{i=1}^{d} p_i \ln p_i = S_{\text{ABB}}[\tau], \tag{42}$$

we find that the ABB entropy of a transition matrix equals the logarithm of the rank of the isometric map distilled from its infinite copies, normalized by the number of copies.

We can give another interpretation of ABB entropy of $\tau$ based on the fact that it equals the von Neumann entropy of the output state $\rho_\tau^{\max}$ in (7). This entropy equality implies that the transition matrix $\tilde{\tau}^{\otimes m}$ inherits the distillation process from $(\rho_\tau^{\max})^{\otimes m}$. Based on (7), we have the expansion

$$(\rho_\tau^{\max})^{\otimes m} = \sum_{\mathbf{k}} P_{\mathbf{k}} \rho_{\mathbf{k}}^{\max}, \quad \rho_{\mathbf{k}}^{\max} = \frac{1}{d_{\mathbf{k}}} \sum_{\mu=1}^{d_{\mathbf{k}}} |\tilde{C}_\mu^{\mathbf{k}}\rangle\langle\tilde{C}_\mu^{\mathbf{k}}|, \tag{43}$$

where $\rho_{\mathbf{k}}^{\max}$ is the rank-$d_{\mathbf{k}}$ MMS. The probability of collapsing into $\rho_{\mathbf{k}}^{\max}$ is also $P_{\mathbf{k}}$, which is consistent with (41) and results in the same story of concentration on $\rho_{\mathbf{k}^*}^{\max}$ in the large $m$-limit. Thus, based on (42) again, the ABB entropy of a transition matrix also equals the logarithm of the rank of the MMS distilled from the infinite copies of the output state of the transition matrix given a full-rank maximally mixed input, normalized by the number of copies.

We showed that the ABB entropy of a transition matrix admits a probabilistic interpretation, where the probability in the diagonal case equals the joint probability of entanglement distillation from the constituent states, modulo suppression by post-selection. Next, we will show that the (modified) pseudo and SVD entropies do not admit such a probabilistic interpretation.

### 3.2.2 Pseudo entropy

To compare with the distillation of $\tilde{\tau}$ for the ABB entropy, we first revisit the distillation of $\hat{\tau}$ for the (modified) pseudo entropy presented in [16] and then comment on its lack of probabilistic interpretation. As discussed in Sec. 2.2.2, the Schmidt bases $\{|\gamma_i\rangle\}$, $\{|\alpha_j\rangle\}$ are not mutually orthogonal, rendering them unsuitable for direct probabilistic interpretation. To analyze the concentration behavior of $\hat{\tau}^{\otimes m}$, we instead use the diagonalized form of $\hat{\tau}$ as given in (13), expressed in terms of potentially complex coefficients $\{\lambda_i\}$ and a pair of bi-orthogonal bases $\{|r_i\rangle\}$, $\{|l_i\rangle\}$. Following [16], the $m$ copies $\hat{\tau}^{\otimes m}$ can be expanded as

$$\hat{\tau}^{\otimes m} = \sum_{\mathbf{k}} \Psi_{\mathbf{k}} V_{\mathbf{k}}, \quad \Psi_k = d_{\mathbf{k}} \prod_{i=1}^{d} \lambda_i^{k_i}, \quad V_{\mathbf{k}} = \frac{1}{d_{\mathbf{k}}} \sum_{\mu=1}^{d_{\mathbf{k}}} |R_\mu^{\mathbf{k}}\rangle \langle L_\mu^{\mathbf{k}}|, \tag{44}$$

with $\sum_{\mathbf{k}} \Psi_{\mathbf{k}} = 1$, $|R_\mu^{\mathbf{k}}\rangle$ the tensor product of $\{|r_i\rangle\}$, $|L_\mu^{\mathbf{k}}\rangle$ the tensor product of $\{|l_i\rangle\}$, and $\mathrm{Tr}\, V_{\mathbf{k}} = 1$. Thanks for the bi-orthogonality $\langle L_\mu^{\mathbf{k}}|R_\nu^{\mathbf{l}}\rangle = \delta_{\mathbf{kl}}\delta_{\mu\nu}$, although the sub transition matrix $V_{\mathbf{k}}$ is non-Hermitian in general, its von Neumman entropy coincides with the logarithm of its rank, namely $S_{\mathrm{von}}[V_{\mathbf{k}}] = \ln d_{\mathbf{k}}$.

In general, $\lambda_i$ takes a complex value, so we cannot identify $\{\Psi_{\mathbf{k}}\}$ as a probability distribution. We may consider the special case of $\lambda_i \geq 0$, $\forall i$ such that $\Psi_{\mathbf{k}} \geq 0, \forall \mathbf{k}$. In this case, the maximum of $\Psi_{\mathbf{k}}$ is attained at $\mathbf{k}^* = (m\lambda_1, \cdots, m\lambda_d)$. In the large-$m$ limit, we observe that the pseudo entropy of $\hat{\tau}^{\otimes m}$ coincides with $S_{\mathrm{von}}[V_{\mathbf{k}^*}]$:

$$\lim_{m\to\infty} \frac{\ln d_{\mathbf{k}^*}}{m} = -\sum_{i=1}^{d} \lambda_i \ln \lambda_i = S_{\mathrm{P}}[\tau], \tag{45}$$

similar to the relation in ABB entropy (42), but their probabilistic interpretations are different. Although the expansion in (44) resembles that of a density matrix decomposition like (43), the former is an expansion of transition matrices, while the latter is an expansion of density matrices. Consequently, the coefficients $\Psi_{\mathbf{k}}$ should be interpreted as summations of amplitudes rather than probabilities even they are non-negative in this special case. In fact, we have

$$\Psi_{\mathbf{k}} = \sum_{\mu=1}^{d_{\mathbf{k}}} \left\langle L_{\mu}^{\mathbf{k}} \right| \hat{\tau}^{\otimes m} \left| R_{\mu}^{\mathbf{k}} \right\rangle \overset{\text{diag.}}{=} \frac{\sqrt{P_{1\mathbf{k}} P_{2\mathbf{k}}}}{\sum_{\mathbf{j}} \sqrt{P_{1\mathbf{j}} P_{2\mathbf{j}}}}. \tag{46}$$

In the last step, we consider the diagonal case together with the condition of Hermitian and positive semi-definite $\hat{\tau}$, namely $|\gamma_i\rangle = |\tilde{\gamma}_i\rangle = e^{i\theta} |\tilde{\alpha}_i\rangle = e^{i\theta} |\alpha_i\rangle$, such that $\Psi_{\mathbf{k}}$ is expressed as the normalized *square root* of the joint probability $P_{1\mathbf{k}} P_{2\mathbf{k}}$, which further emphasizes that $\Psi_k$ is not a conventional probability in the quantum mechanical sense. Furthermore, even $|\Psi_{\mathbf{k}}|^2$ cannot be interpreted as a physical probability, despite being linear in $P_{1\mathbf{k}} P_{2\mathbf{k}}$ and sharing the same extreme point $\mathbf{k}^*$, since it represents the square of the summation of a series of amplitudes rather than a square of single amplitude,

$$|\Psi_{\mathbf{k}}|^2 = \left| \sum_{\mu=1}^{d_{\mathbf{k}}} \left\langle L_{\mu}^{\mathbf{k}} \right| \hat{\tau}^{\otimes m} \left| R_{\mu}^{\mathbf{k}} \right\rangle \right|^2. \tag{47}$$

### 3.2.3 SVD entropy

Finally we revisit the distillation of $\bar{\tau}$ for the SVD entropy presented in [67] and then comment on its lack of probabilistic interpretation, which is similar to the discussion on pseudo entropy.

Following [67], we consider the $m$ copies of $\bar{\tau}$ and expand $\bar{\tau}^{\otimes m}$ as

$$\bar{\tau}^{\otimes m} = \sum_{\mathbf{k}} \Phi_{\mathbf{k}} W_{\mathbf{k}}, \quad \Phi_{\mathbf{k}} = d_{\mathbf{k}} \prod_{i=1}^{d} q_i^{k_i}, \quad W_{\mathbf{k}} = \frac{1}{d_{\mathbf{k}}} \sum_{\mu=1}^{d_{\mathbf{k}}} \left| \tilde{A}_{\mu}^{\mathbf{k}} \right\rangle \left\langle \tilde{A}_{\mu}^{\mathbf{k}} \right|. \tag{48}$$

where $W_{\mathbf{k}}$ is a $d_{\mathbf{k}}$-dimensional transition matrix with $\operatorname{Tr} W_{\mathbf{k}} = 1$, and the coefficients $\{\Phi_{\mathbf{k}}\}$ form a normalized distribution, $\sum_{\mathbf{k}} \Phi_{\mathbf{k}} = 1$. In [67], both $\bar{\tau}^{\otimes m}$ and each $W_{\mathbf{k}}$ are treated formally as density matrices, with $\{\Phi_{\mathbf{k}}\}$ interpreted as a probability distribution directly. The maximum of this distribution is attained at $\mathbf{k}^* = (mq_1, \cdots, mq_d)$, and the SVD entropy of $\bar{\tau}^{\otimes m}$ is identified with the von Neumann entropy of $W_{\mathbf{k}^*}$ in the large-$m$ limit

$$\lim_{m \to \infty} \frac{\ln d_{\mathbf{k}^*}}{m} = -\sum_{i=1}^{d} q_i \ln q_i = S_{\text{SVD}}[\tau]. \tag{49}$$

However, $\bar{\tau}^{\otimes m}$ and $W_{\mathbf{k}}$ should be interpreted as transition matrices rather than density matrices, due to the presence of the square root in the construction of $\bar{\tau}$ in (18). This holds even though they are formally Hermitian, positive semi-definite, and normalized to unit trace. Thus, the decomposition of (48) is still an expansion of transition matrices. Consequently, the coefficients $\Phi_{\mathbf{k}}$ should be still interpreted as summations of amplitudes rather than probabilities. In fact, we have

$$\Phi_{\mathbf{k}} = \sum_{\mu=1}^{d_{\mathbf{k}}} \left\langle \tilde{A}_{\mu}^{\mathbf{k}} \right| \bar{\tau}^{\otimes m} \left| \tilde{A}_{\mu}^{\mathbf{k}} \right\rangle = d_{\mathbf{k}} \prod_{i=1}^{d} q_i^{k_i} \overset{\text{diag.}}{=} \frac{\sqrt{P_{1\mathbf{k}} P_{2\mathbf{k}}}}{\sum_{\mathbf{j}} \sqrt{P_{1\mathbf{j}} P_{2\mathbf{j}}}}. \tag{50}$$

In the last step, $\Phi_{\mathbf{k}}$ is given by the normalized square root of the joint probability $P_{1\mathbf{k}}P_{2\mathbf{k}}$, which again underscores that $\Phi_k$ does not represent a conventional quantum-mechanical probability. In the same way, $|\Phi_{\mathbf{k}}|^2$ cannot be interpreted as a physical probability, despite being linear in $P_{1\mathbf{k}}P_{2\mathbf{k}}$ and sharing the same extreme point $\mathbf{k}^*$, since it represents the square of the summation of a series of amplitudes rather than a square of single amplitude,

$$|\Phi_{\mathbf{k}}|^2 = \left| \sum_{\mu=1}^{d_{\mathbf{k}}} \left\langle \tilde{A}_\mu^{\mathbf{k}} \right| \bar{\tau}^{\otimes m} \left| \tilde{A}_\mu^{\mathbf{k}} \right\rangle \right|^2 . \tag{51}$$

# 4 Haar random states and Page curves

In this section, we are going to investigate the universal behaviors of various entropy measures for Haar random states [92]. A Haar random state is generated by applying a unitary matrix, drawn from the circular unitary ensemble (CUE), to a fixed reference state. Due to the unitary invariance of the Gaussian unitary ensemble (GUE) measure, the eigenvectors of a GUE matrix are also distributed according to the Haar measure. Therefore, any single eigenvector of a GUE matrix is a Haar random state [93–95]. The average subsystem entanglement entropy of Haar random states exhibits a universal behavior as a function of subsystem size, following the expression $\langle S \rangle \approx \ln d_a - d_a/2d_b$, under the condition of $1 \ll d_a \leq d_b$, where $d_a$ and $d_b$ represent the Hilbert space dimensions of the subsystems with the total dimension being $D = d_a d_b$. For the opposite case $d_b \leq d_a$, the formula applies with $d_a$ and $d_b$ swapped. As a function of the subsystem size $\ln d_a$ at fixed total Hilbert space dimension $D$, the entropy increases linearly, reaches a maximum near balanced subsystem sizes, and then decreases linearly, forming the so-called Page curve [72, 73, 96].

The behavior of the pseudo entropy over two independent Haar random states has been explored in [16], where it was shown that its complex-valued distribution is notably extensive. This is in contrast to the explicit concentration of entanglement entropy for a single Haar random state around the logarithm of Hilbert space dimension. The SVD entropy was shown to exhibit a behavior analogous to the Page curve in [67].

In this section, we evaluate the averages of the modified pseudo entropy and ABB entropy over the Haar random ensemble, as well as the pseudo entropy and SVD entropy, and also compare their properties. We consider two independent Haar random states, $|\psi_1\rangle_{bc}$ and $|\psi_2\rangle_{ab}$, and the resulting transition matrix $\tau$ is generally a non-square matrix. In the large dimension limit, with fixed proportion $\ln d_a : \ln d_b : \ln d_c$, we first demonstrate analytically that both the SVD and ABB entropies are dominated by the logarithm of the smallest Hilbert space dimension. We then calculate the averages of all four types of entropies for different subsystem sizes numerically.

As demonstrated in [97], the ensemble-averaged Rényi entropy for a density matrix $\rho$ admits the approximation

$$\overline{S_n[\rho]} = \frac{1}{1-n} \overline{\ln \frac{\text{Tr}\,[\rho^n]}{\text{Tr}\,[\rho]^n}} \approx \frac{1}{1-n} \ln \frac{\overline{\text{Tr}\,[\rho^n]}}{\overline{\text{Tr}\,[\rho]^n}}, \tag{52}$$

where the fluctuation term is significantly suppressed in the large-dimension limit. Consequently, the Haar random average of the Rényi versions of the SVD entropy (21) and ABB entropy (11) reduces to evaluating the quantities,

$$\left\langle \text{Tr}\left[\left(\tau^\dagger \tau\right)^n\right]\right\rangle_{\text{Haar}}, \ \left\langle \text{Tr}[\tau^\dagger \tau]^n\right\rangle_{\text{Haar}}, \ \left\langle \text{Tr}[\left(\tau^\dagger \tau\right)^{\frac{mn}{2}}]\right\rangle_{\text{Haar}}, \ \left\langle \text{Tr}[\left(\tau^\dagger \tau\right)^{\frac{m}{2}}]^n\right\rangle_{\text{Haar}}. \tag{53}$$

We construct two Haar random states by $|\psi_1\rangle_{bc} = U_{bc} |0\rangle_{bc}$ and $|\psi_2\rangle_{ab} = V_{ab} |0\rangle_{ab}$, where $U_{bc}$ and $V_{ab}$, sampled independently from two CUEs, are two unitary matrices acting on $\mathcal{H}_{bc}$ and $\mathcal{H}_{ab}$, respectively. The transition matrix is then given by

$$\tau = \mathrm{Tr}_b \left[ U_{bc} |0\rangle_{bc} \langle 0|_{ab} V_{ab}^\dagger \right]. \tag{54}$$

Using this, the Haar random average of one of the required traces can be expressed as

$$\left\langle \mathrm{Tr} \left[ (\tau^\dagger \tau)^n \right] \right\rangle_{\mathrm{Haar}} = \mathrm{Tr} \int_{U,V} \left( \mathrm{Tr}_b \left[ V_{ab} |0\rangle_{ab} \langle 0|_{bc} U_{bc}^\dagger \right] \mathrm{Tr}_b \left[ U_{bc} |0\rangle_{bc} \langle 0|_{ab} V_{ab}^\dagger \right] \right)^n dU \, dV. \tag{55}$$

To perform the ensemble averages, we use a key result derived from Schur's lemma in [98,99], which states that for a $D$-dimensional Haar random state $|\psi\rangle = U |0\rangle$,

$$\int_{\mathrm{Haar}} (|\psi\rangle \langle \psi|)^{\otimes k} \, d\psi = \frac{\sum_{\pi \in S_k} P_\pi}{D(D+1) \cdots (D+k-1)}, \tag{56}$$

where $S_k$ denotes the permutation group of order $|S_k| = k!$, and $P_\pi$ is the permutation operator with $\pi = \pi(1) \cdots \pi(k)$, defined as $P_\pi |i_1, \cdots, i_k\rangle = |i_{\pi(1)}, \cdots, i_{\pi(k)}\rangle$. For the special case with $n = 2$ and $m = 2$, where $S_{\mathrm{SVD}}^{(2,2)}[\tau]$ coincides with $S_{\mathrm{ABB}}^{(2)}[\tau]$, we employ Wick's theorem and the result (56) for $k = 1$ to obtain the averages as

$$\left\langle \mathrm{Tr} \left[ (\tau^\dagger \tau)^2 \right] \right\rangle_{\mathrm{Haar}} = \frac{d_a d_b^2 d_c^2 + d_a^2 d_b d_c^2 + d_a^2 d_b^2 d_c}{d_a d_b^2 d_c (d_a d_b + 1)(d_b d_c + 1)}, \tag{57}$$

$$\left\langle \mathrm{Tr} \left[ \tau^\dagger \tau \right]^2 \right\rangle_{\mathrm{Haar}} = \frac{d_a^2 d_b^2 d_c^2 + d_a^2 d_b d_c + d_a d_b^2 d_c + d_a d_b d_c^2}{d_a d_b^2 d_c (d_a d_b + 1)(d_b d_c + 1)}. \tag{58}$$

The corresponding average of the Rényi ABB entropy for $n = 2$ is thus given by

$$\left\langle S_{\mathrm{ABB}}^{(2)}[\tau] \right\rangle_{\mathrm{Haar}} = -\ln \frac{d_a d_b^2 d_c^2 + d_a^2 d_b d_c^2 + d_a^2 d_b^2 d_c}{d_a^2 d_b^2 d_c^2 + d_a^2 d_b d_c + d_a d_b^2 d_c + d_a d_b d_c^2}. \tag{59}$$

For general $n$ and $m$, the Haar random averages in (53) may be evaluated by combining the permutation operator method [97], Wick's theorem and the result (56), or alternatively, by utilizing the Weingarten functions as in [100–102]. However, exact expressions become increasingly cumbersome as $n$ and $m$ grow. We therefore restrict our presentation to the leading-order contributions

$$\left\langle S_{\mathrm{SVD}}^{(n,m)}[\tau] \right\rangle_{\mathrm{Haar}} = \frac{1}{1-n} \ln \frac{d_a d_b^{\frac{mn}{2}} d_c^{\frac{mn}{2}} + d_a^{\frac{mn}{2}} d_b d_c^{\frac{mn}{2}} + d_a^{\frac{mn}{2}} d_b^{\frac{mn}{2}} d_c + \cdots}{d_a^n d_b^{\frac{mn}{2}} d_c^{\frac{mn}{2}} + d_a^{\frac{mn}{2}} d_b^n d_c^{\frac{mn}{2}} + d_a^{\frac{mn}{2}} d_b^{\frac{mn}{2}} d_c^n + \cdots}, \tag{60}$$

where we consider $n > 1$ and $m \geq 2$ since the analytical continuation starts from even $m$. A similar analysis yields the dominant term of the $n$-Rényi ABB entropy

$$\left\langle S_{\mathrm{ABB}}^{(n)}[\tau] \right\rangle_{\mathrm{Haar}} = \frac{1}{1-n} \ln \frac{d_a d_b^n d_c^n + d_a^n d_b d_c^n + d_a^n d_b^n d_c + \cdots}{d_a^n d_b^n d_c^n + \cdots}, \tag{61}$$

where we consider $n > 1$. In the limit of large dimensions with fixed proportion $\ln d_a : \ln d_b : \ln d_c$, both expressions, (60) and (61) reduce to $\ln \min [d_a, d_b, d_c]$, implying that the leading-order terms of the averaged Rényi SVD and ABB entropy concentrate around the

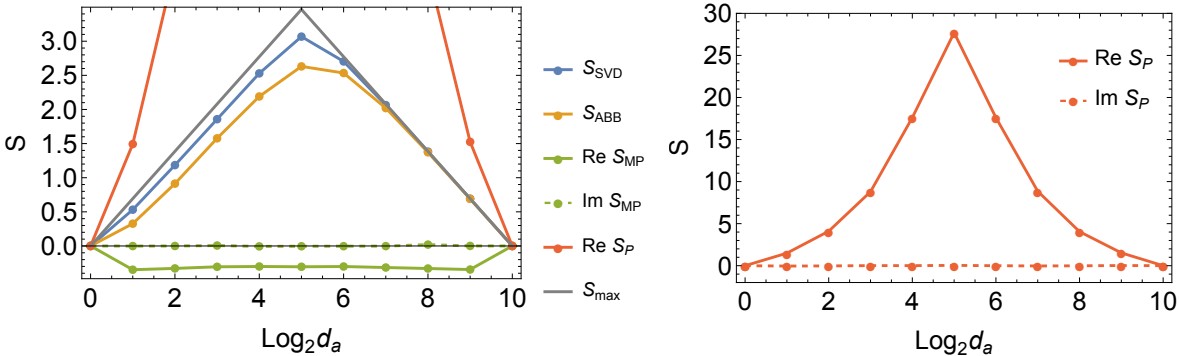

Figure 5: Left: Haar random averages for the SVD, ABB and (modified) pseudo entropy, excluding the imaginary part of pseudo entropy. The gray curve denotes the leading-order term given by (62). Right: Real and imaginary parts of the pseudo entropy. Numerical results are performed over 100 disorder realizations for the SVD and ABB entropy, and $5 \times 10^4$ disorder realizations for the (modified) pseudo entropy, under the condition $d_a = d_c$ and $D = d_a d_b = 2^{10}$.

logarithm of the smallest subsystem's Hilbert space dimension, independent of the Rényi index $n$. Thus, the averages for the SVD and ABB entropy are respectively dominated by

$$\langle S_{\text{SVD}}[\tau] \rangle = \ln \min [d_a, d_b, d_c] + \cdots, \quad \langle S_{\text{ABB}}[\tau] \rangle = \ln \min [d_a, d_b, d_c] + \cdots, \tag{62}$$

where the next-to-leading terms for the averages of two entropies are different.

When identifying $a$ and $c$, we can consider the (modified) pseudo entropy of $\tau$. Its conjugate transpose, $\tau^\dagger = \text{Tr}_b[V_{ab} |0\rangle_{ab} \langle 0|_{ab} U_{ab}^\dagger]$, is generated from two random states and belongs to the same ensemble as $\tau$, with equal statistical weight. Since the (modified) pseudo entropy of $\tau^\dagger$ is the complex conjugate of that of $\tau$, the ensemble average of the (modified) pseudo entropy is therefore expected to be real.

We numerically calculate the ensemble averages of all four types of entropy for varying subsystem sizes $\log_2 d_a$ under the condition $d_a = d_c$ and $d_a d_b = D = $ const., as shown in Fig. 5. Our results reproduce the behavior of the average pseudo entropy reported in [16], confirming that the ensemble average of $\text{Re} \, S_P[\tau]$ remains positive while the imaginary part vanishes upon averaging. We find that although the real part exhibits a similar Page curve behavior, it significantly exceeds $\ln d_a$, the maximal value of the entanglement entropy in a Hilbert space of the same dimension. This phenomenon arises from the non-Hermitian nature of the transition matrix $\hat{\tau}$.

To further investigate the origin of this behavior, we analyze the distribution of the spectrum $\{\lambda_i\}$ of $\hat{\tau}$ in the Haar random ensemble, as shown in Fig. 6. We observe that the spectral distribution exhibits approximate rotational symmetry centered around $\bar{\lambda} = 1/d_a$, a consequence of the normalization condition $\text{Tr} \, \hat{\tau} = \sum_i^{d_a} \lambda_i = 1$. Given that $\bar{\lambda}$ lies on the real axis, the rotational symmetry of the spectrum is consistent with the vanishing of $\text{Im} \, S_P[\tau]$ in the ensemble average. The function $\text{Re}[-z \ln z]$ is positive over more than half of the complex plane, as illustrated in the left panel of Fig. 2. Although the rotationally invariant spectral distribution is slightly shifted along the positive real axis, the majority of the spectrum still lies within the region where $\text{Re}[-z \ln z]$ is positive. Consequently, the ensemble-averaged pseudo entropy $S_P$ typically acquires a large positive value for the spectral distribution observed in Fig. 6.

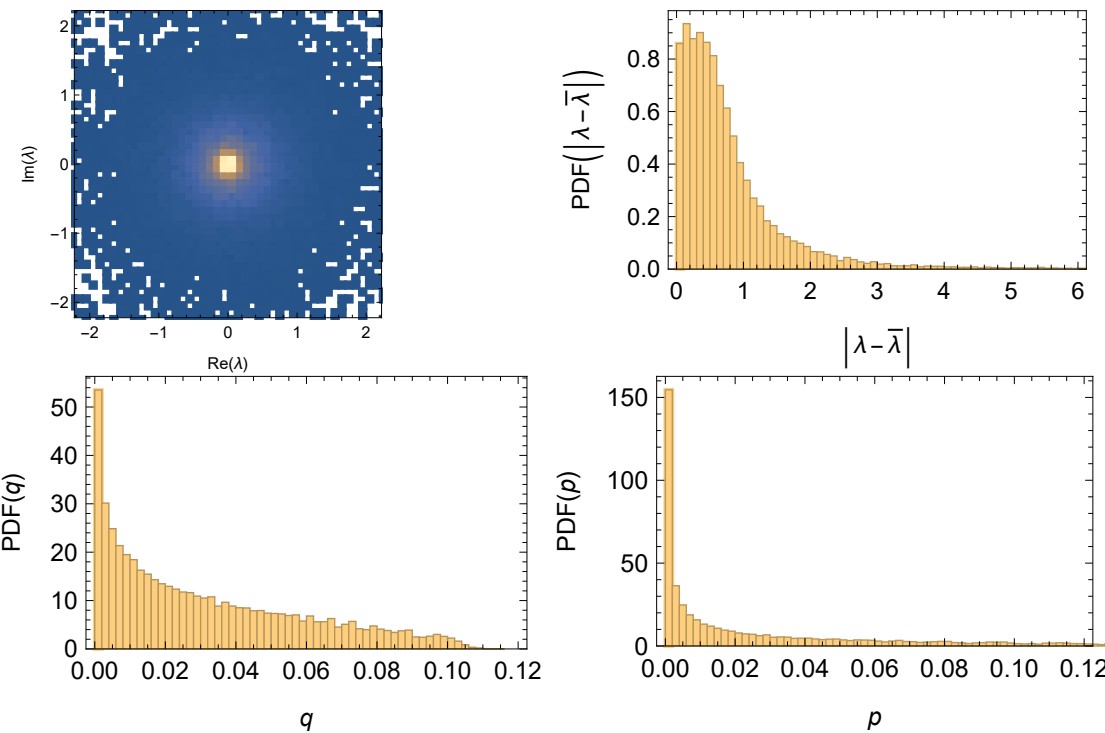

Figure 6: The probability density functions (PDFs) of the eigenvalues $\{\lambda_i\}$, $\{|\lambda_i - \bar{\lambda}|\}$, $\{q_i\}$, and $\{p_i\}$ for (modified) pseudo entropy, SVD entropy, and ABB entropy in 1000 pairs of Haar random realizations at $d_a = d_b = d_c = 2^5$.

The ensemble-averaged imaginary part of the modified pseudo entropy $\text{Im}\, S_{\text{MP}}[\tau]$ also vanishes, similarly to the pseudo entropy. However, in contrast to the pseudo entropy, the real part $\text{Re}\, S_{\text{MP}}[\tau]$ is significantly suppressed and form a stable plateau around a negative value of approximately $-0.3$, independent of different subsystem sizes, even though $S_{\text{P}}$ and $S_{\text{MP}}$ are derived from the same spectral distribution. Furthermore, we confirm that this negative plateau persists in various total system sizes, including $D = 128$, $256$, $512$, and $1024$.

As shown in the right panel of Fig. 2, the function $\text{Re}[-z \ln |z|]$ is antisymmetric with respect to the imaginary axis. It takes negative values in the right half complex plane ($\text{Re}\, z > 0$), and positive values in the left half plane ($\text{Re}\, z < 0$), except within the unit disk ($|z| \leq 1$), where the sign is reversed. Due to the slight shift of the spectral distribution toward the positive real axis, the portion of the spectrum inside the unit disk contributes positively to $\text{Re}\, S_{\text{MP}}$. However, the spectrum also has a considerable weight outside this unit disk (see PDF of $|\lambda - \bar{\lambda}|$ in Fig. 6), over which the average value of $\text{Re}\, S_{\text{MP}}$ is negative. As a whole, the net negative contribution from outside the unit disk dominates over the positive contribution inside the unit disk. Thus, the ensemble-averaged $\text{Re}\, S_{\text{MP}}$ acquires a small negative value.

Finally, both the ensemble-averaged SVD and ABB entropy exhibit sharp concentration around $\ln d_a$, analogous to the Page curve of Haar random states. This is consistent with the analytical prediction (62), arises because the spectra $\{q_i\}$ of $\bar{\tau}$ and $\{p_i\}$ of $\tilde{\tau}\tilde{\tau}^\dagger$ are confined to the interval $[0, 1]$, just as in the case of conventional entanglement entropy. According to (62), the entropy curves are symmetric under permutations of $d_a$, $d_b$ and $d_c$. However, this symmetry is absent when two dimensions are equal, e.g., $d_a = d_c$, and one of $\{d_a, d_b\}$ remains finite. In such a case, the SVD and ABB entropies are no longer

symmetric with respect to $d_a \leftrightarrow d_b$, a feature clearly illustrated in Fig. 5. When both $d_a$ and $d_b$ become sufficiently large, the symmetry is approximately restored, which can be verified in the case of $n = 2$ Rényi ABB entropy in Eq. (59). For $d_a < d_b$, the SVD entropy always exceeds the ABB entropy, with both peaking around $d_a = d_b$. However, beyond this point, the two entropies gradually converge as $d_a$ increases. This behavior can be explained by their spectral distributions. As illustrated in Fig. 6, the distribution of $\{p_i\}$ is more concentrated than $\{q_i\}$ due to the quadratic relation $p_i \propto q_i^2$, resulting in a lower ABB entropy compared to the SVD entropy.

Thus, we conclude that only the SVD and ABB entropies of the transition matrix, constructed from two independent Haar random states, approach the Page curve in the large-dimension limit, consistent with the subsystem entanglement entropy of a single random state. By contrast, the (modified) pseudo entropy exhibits substantial deviations.

# 5 Bi-orthogonal eigenstates of non-Hermitian random systems

In the previous section, we examined the behavior of ensemble averages for all four types of entropy associated with the transition matrix $\tau$, constructed from two independent random states. In this section, we turn to the transition matrices constructed from two correlated random states $\langle \psi_1 | = \langle L_n |$, $|\psi_2\rangle = |R_n\rangle$, namely

$$\tau = \text{Tr}_b |R_n\rangle \langle L_n|, \tag{63}$$

where $\{\langle L_n |, |R_n\rangle\}$ are taken from the bi-orthogonal basis [49] of a non-Hermitian matrix $H$ of systems $a$ and $b$ defined by

$$H|R_n\rangle = \mathcal{E}_n |R_n\rangle, \quad \langle L_n| H = \mathcal{E}_n \langle L_n|, \quad \mathcal{E}_n \in \mathbb{C} \tag{64}$$

with $\langle L_m | R_n \rangle = \delta_{mn}$, and $\text{Tr}_b$ is the partial trace on a subsystem $b$.

We consider the non-Hermitian Hamiltonian $H$ drawn from the Ginibre unitary ensemble (GinUE) or the non-Hermitian SYK model. The modified pseudo entropies of such kind of transition matrices, were computed in [63], revealing a significant suppression of positive values compared to the Page curve of entanglement entropy. Following the analysis in the previous section, we investigate the origin of the plateau values in the bi-orthogonal case by examining the spectral distribution in the ensemble. In this context, we further compute the ensemble averages of the pseudo entropy, ABB entropy, and SVD entropy for varying subsystem sizes and analyze their behaviors from the spectral distribution. Besides, we also explore the universality of our results by analyzing the non-Hermitian SYK model. We will see that the ABB and SVD entropy of the transition matrices constructed from bi-orthogonal random eigenstates exhibit the behavior of the Page curve, in contrast to the plateau behavior of the modified pseudo entropy discovered in [63].

## 5.1 Ginibre unitary ensemble

The Hamiltonian of dimension $D \times D$ in the GinUE [103] is defined as $H = \frac{1}{\sqrt{2}} (H_1 + iH_2)$, where $H_1$ and $H_2$ are real matrices whose entries are independently sampled from the Gaussian distribution with zero mean value and variance $D^{-1}$. Here, "unitary" refers to

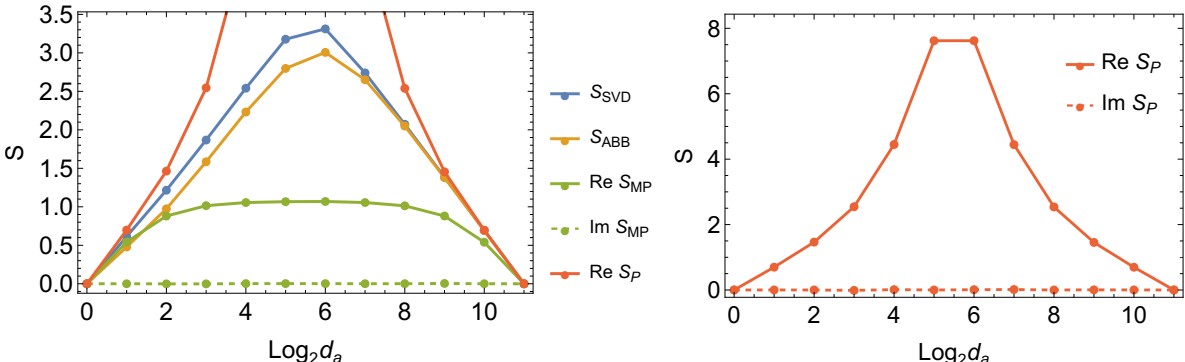

Figure 7: The ensemble averages of the (modified) pseudo entropy, SVD entropy, and ABB entropy for different bi-orthogonal eigenstates of the GinUE with $D = 2^{11}$. The corresponding eigenvalues are $w = |w| e^{i\theta}$ and $|w| = 0.9$, with $\theta = k\pi/16$, $k \in 0, \cdots, 31$, and we perform the average over 256 disorder realizations.

the bi-unitary invariance of the probability density distribution of $H$ under the transformation $H \to UHV$, with $U$ and $V$ being two arbitrary unitary matrices [104]. The GinUE is identified as class A in the Altland-Zirnbauer (AZ) classification [105].

It is well established [103, 106] that in the GinUE, the eigenvalue density follows the circular law in the limit $D \to \infty$; that is, the eigenvalues become uniformly distributed within the unit disk in the complex plane:

$$\rho(w) = \frac{1}{\pi}, \quad \text{for } |w| \leq 1, \quad \text{and otherwise,} \quad \rho(w) = 0, \quad \text{for } |w| > 1.$$

In the large $D$ limit, the spectrum becomes so dense that, given a complex number $w$ with $|w| \leq 1$, one can always find eigenvalues arbitrarily close to $w$.

At finite $D$, however, the spectrum is discrete and $w$ need not coincide with any eigenvalue. To identify the closest one, let $n_w$ label the bi-orthogonal eigenstates $\{\langle L_{n_w}|, |R_{n_w}\rangle\}$ whose eigenvalue $\mathcal{E}_{n_w}$ is closest to $w$ in the complex plane. They could be computed by applying the Arnoldi method [107] to the shifted Hamiltonian $H - w$ and selecting the bi-orthogonal eigenstates of smallest absolute eigenvalue. The transition matrix $\tau$ is then constructed via Eq. (63).

Because $H$ is drawn from the GinUE ensemble, the states $|R_{n_w}\rangle$ and $|L_{n_w}\rangle$ are correlated Haar random states, unlike the situation in Sec. 4, where $\tau$ was constructed from two independent Haar random states.

Similar to the previous section, since $H$ and $H^\dagger$ belong to the same GinUE ensemble, the corresponding transition matrices $\tau$ and $\tau^\dagger$ are drawn from the same ensemble. As a result, both $\text{Im}\,S_P$ and $\text{Im}\,S_{MP}$ vanish upon ensemble averaging. As shown in Fig. 7, we find that $\text{Re}\,S_P$ in the ensemble average is also greatly enhanced, since the spectral distribution of $\hat{\tau}$, shown in Fig. 8, is largely concentrated in the region where $\text{Re}[-z \ln z]$ remains positive, while the portion of the spectrum lying in the region where $\text{Re}[-z \ln z]$ is negative is very sparse, as illustrated in Fig. 2.

The real part of the modified pseudo entropy $\text{Re}\,S_{MP}$ was shown in [63] to be significantly suppressed, forming a plateau in the ensemble average. The plateau value was analytically obtained as

$$\langle S_{MP} \rangle = \frac{1 - \gamma - \ln(1 - |w|^2)}{2}, \tag{65}$$

in the limit of $1 \ll d_a \ll d_b$, where $\gamma$ is the Euler-Mascheroni constant. Obviously, this expression is positive and depends only on $|w|$, but remains independent of subsystem dimension $d_a$ and total dimension $D$. As shown by the green curve in Fig. 7, our numerical results for the modified pseudo entropy at $D = 2^{11}$ reproduce the plateau at $D = 2^{14}$ reported in [63]. In both cases, they match well with this analytical expression (65) even at $d_a = d_b$. To further investigate the origin of the plateau values in the GinUE setting, we analyze the spectral distribution of $\hat{\tau}$ again.

As shown in the upper left panel of Fig. 8, the spectral distribution is also approximately rotationally invariant around the center $\bar{\lambda} = 1/d_a$. The shift of the center from the origin by $\bar{\lambda}$ results in $\operatorname{Re} S_{\mathrm{MP}}$ receiving greater positive contributions from the right half of the unit disk ($\operatorname{Re} z > 0$ and $|z| \leq 1$) than negative contributions from the left half ($\operatorname{Re} z < 0$ and $|z| \leq 1$), yielding a positive but suppressed value of $\operatorname{Re} S_{\mathrm{MP}}$ in the ensemble average. However, unlike the case in Fig. 6, the distribution is concentrated within the disk with a radius less than one. As shown in the upper right panel of Fig. 8, the probability density distribution of $|\lambda - \bar{\lambda}|$ peaks in this small region and decays rapidly, approaching zero around $|\lambda - \bar{\lambda}| \approx 0.8$. This agrees well with the asymptotic result in the large-$D$ limit and under the condition $1 \ll d_a \ll d_b$, as given in [63],

$$\rho(x) = 2x \left( 2 - e^{-x^{-2}} \left( 2 + 2x^{-2} + x^{-4} \right) \right), \quad \text{with } x = \frac{|\lambda - \bar{\lambda}|}{\sqrt{1 - |w|^2}}. \tag{66}$$

Consequently, $\operatorname{Re} S_{\mathrm{MP}}$ remains suppressed but positive, although there are negative contributions from outside the unit disk. Its value also depends on $|w|$, as the spectral distribution of $\hat{\tau}$ is itself a function of $|w|$.

We further numerically calculate the ensemble averages of the SVD entropy and ABB entropy for $\tau$ in (63), with the results presented in Fig. 7. The ensemble-averaged SVD and ABB entropy exhibit the same behavior as in the case of two independent Haar random states discussed in Sec. 4. In particular, they are independent of the specific eigenvalue $w$. As shown in Fig. 8, the distribution of the spectra $\{q_i\}$ and $\{p_i\}$ lies within the interval $[0, 1]$, with $\{p_i\}$ being much more concentrated than $\{q_i\}$ due to the quadratic relation, so we can still observe that the SVD entropy is greater than the ABB entropy when $d_a < d_b$.

The strong suppression of the modified pseudo entropy of the transition matrix constructed from bi-orthogonal bases has been conjectured to be a universal feature of non-Hermitian chaotic many-body systems [63], in contrast to the universal Page curve observed in Hermitian chaotic systems. However, when the spectrum of the transition matrix is complex, we are even unable to identify a probabilistic interpretation of the (modified) pseudo entropy, as discussed in Sec. 3.2.2, let alone relate it to the entanglement of the individual bi-orthogonal eigenstates. In contrast, the SVD and ABB entropies always exhibit a Page-curve-like behavior, even though the bi-orthogonal eigenstates involved are correlated. We therefore propose taking the Page-curves-like behaviors of the ABB or SVD entropies derived from bi-orthogonal eigenstates as features of non-Hermitian chaotic system, consistent with their use in single eigenstates of Hermitian chaotic systems[3]. We will test this proposal in the context of the non-Hermitian SYK model in the next subsection.

---

[3]Recently, comparisons between the complex-eigenvalue and singular-value spectra of non-Hermitian Hamiltonians and their implications for non-Hermitian chaos have been investigated in [108–113]. Here, we instead focus on the statistics of the entanglement spectrum and the singular-value spectrum of the transition matrix constructed from the bi-orthogonal eigenstates of chaotic and integrable non-Hermitian Hamiltonians.

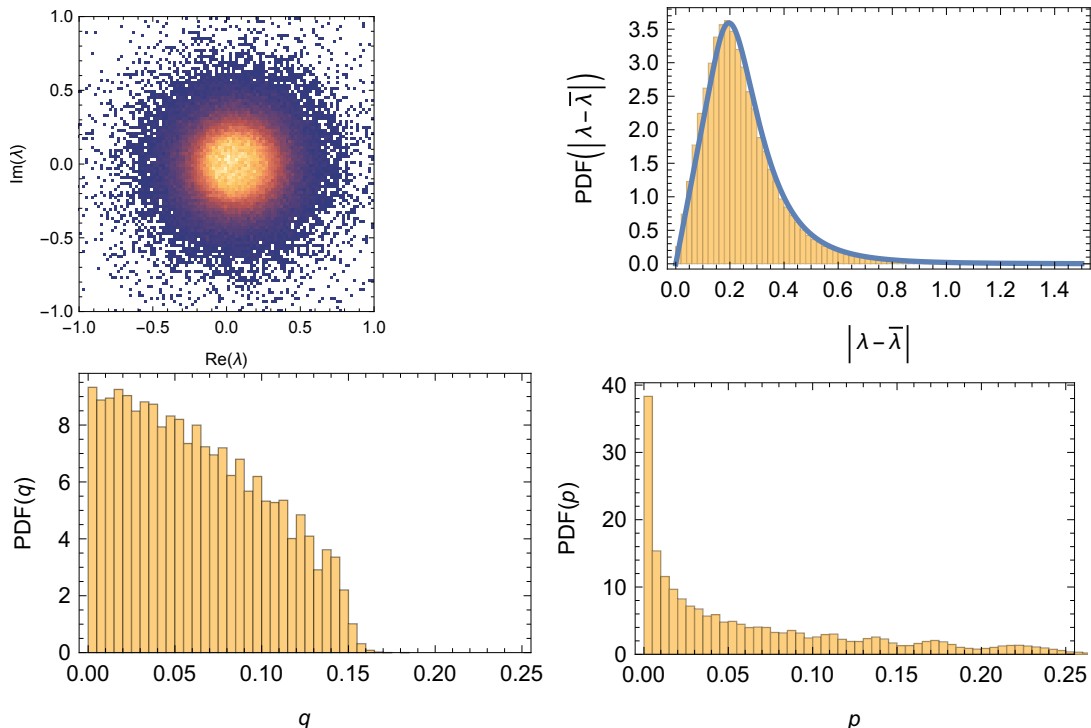

Figure 8: The probability density functions (PDFs) of the eigenvalues $\{\lambda_i\}$, $\left\{\left|\lambda_i - \bar{\lambda}\right|\right\}$, $\{q_i\}$, and $\{p_i\}$ for (modified) pseudo entropy, SVD entropy, and ABB entropy for bi-orthogonal eigenstates of Ginibre matrices at $d_a = 2^4$ and $d_b = 2^7$. We sample from different eigenstates for $w = |w|\,e^{i\theta}$ and $|w| = 0.9$, with $\theta = k\pi/16$, $k \in 0, \cdots, 31$, and also 256 disorder realizations for each case. The blue curve is given by the analytical result of (66).

## 5.2   Non-Hermitian SYK

The Hermitian SYK model consists of $N$ Majorana fermions $\{\psi_i\}_{i=1}^{N}$ interacting through all-to-all $q$-body interactions with real Gaussian-distributed random couplings [114]. For $q = 2$, the model is free and integrable, while for $q \geq 4$, it is chaotic [115]. The entanglement entropy of a subsystem of eigenstates for SYK model was previously studied in [116]. It was shown that when the subsystem size is much smaller than the total system, the entanglement entropy reaches its maximum, i.e., $\ln d_a$. However, for arbitrary $q$, as $d_a$ increases, the value remains below that of Haar random state, i.e., the Page curve. The rigorous proof of this deviation was provided in [117], indicating that the eigenstates of the SYK model do not fully realize the maximal randomness observed in the Haar random states.

The non-Hermitian SYK (nSYK) model is generalized from the Hermitian SYK model by considering complex random couplings, whose Hamiltonian is [118]

$$H_{\text{nSYK}} = \sum_{1 \leq i_1 < \cdots < i_q \leq N} \left( J_{i_1 \cdots i_q} + i M_{i_1 \cdots i_q} \right) \psi_{i_1} \psi_{i_2} \cdots \psi_{i_q}, \tag{67}$$

where $\{\psi_i\}_{i=1}^{N}$ is a set of Majorana fermion operators satisfying $\{\psi_i, \psi_j\} = \delta_{ij}$. The coupling coefficients $J_{i_1 \cdots i_q}$ and $M_{i_1 \cdots i_q}$ are both real Gaussian random variables with zero mean and variance $\frac{J^2(q-1)!}{N^{q-1}}$.

We only consider the nSYK in AZ class A as well. According to [119, 120], it can be realized in nSYK with $N \mod 8 = 2, 4$ and $q \mod 4 = 0, 2$. The eigen spectrum of

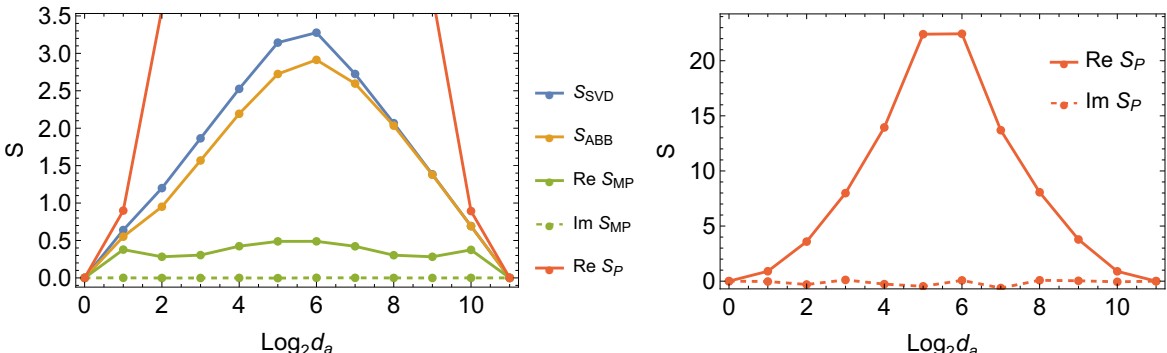

Figure 9: The ensemble averages of the (modified) pseudo entropy, SVD entropy and ABB entropy for the bi-orthogonal eigenstates of the non-Hermitian SYK model with $q = 4$, $N = 22$. The corresponding eigenvalues of bi-orthogonal eigenstates are $w = |w| e^{i\theta}$, where $|w| / |E_{\max}| = 0.73$, and $|E_{\max}|$ denotes the maximal absolute eigenvalue. The phase angles are sampled at $\theta = k\pi/16$, with $k \in 0, \cdots, 31$. For each case, we perform average over 512 disorder realizations.

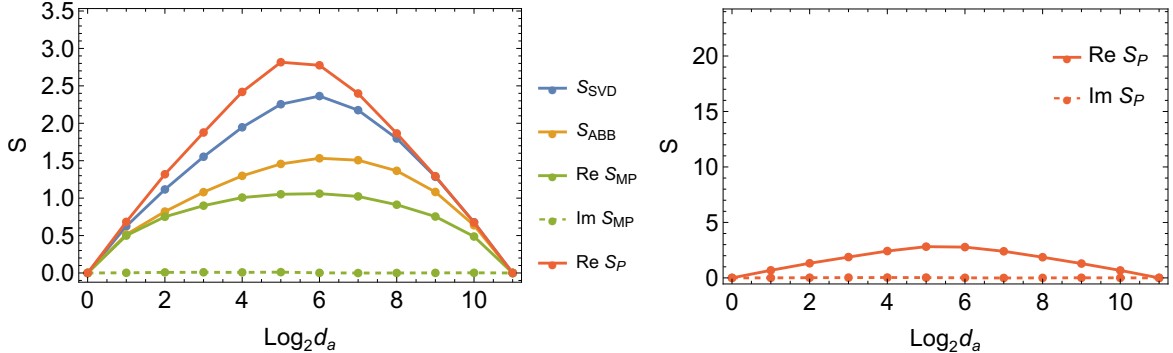

Figure 10: The ensemble averages of the entropies for the bi-orthogonal eigenstates of the non-Hermitian SYK model with $q = 2$, $N = 22$. The corresponding eigenvalues of bi-orthogonal eigenstates are $w = |w| e^{i\theta}$, where $|w| / |E_{\max}| = 0.73$. The phase angles are sampled at $\theta = k\pi/16$, with $k \in 0, \cdots, 31$. For each case, we perform average over 512 disorder realizations.

nSYK exhibits rotational symmetry in the complex plane, but the eigenvalue distribution is non-uniform, distinct from the GinUE [63, 118].

We construct the transition matrix from the bi-orthogonal eigenstates of nSYK and investigate the four types of entropy of the transition matrix. We numerically investigate the nSYK with $N = 22$ and $q = 2, 4$, which still belongs to class A. To compare the entropies between nSYK with different $q$, we choose the bi-orthogonal states of eigenvalue $w$ with $w/ |E_{\max}|$ being the same, where $|E_{\max}|$ is the maximal absolute eigenvalue. Similar to the case of the GinUE, since the ensemble of $H_{\text{nSYK}}$ is the same as its conjugation transpose, the imaginary parts of the averaged entropies vanish, and we focus on their real parts.

At $q = 4$, the averaged entropies as the functions of subsystem size $\log_2 d_a$ are shown in Fig. 9. We observe the following phenomena:

- The pseudo entropy is greatly enhanced.

- The modified pseudo entropy is strongly suppressed and exhibits a plateau, consistent with [63].

- The SVD entropy and the ABB entropy exhibit a growth–peak–decline behavior akin to that seen in the GinUE. These curves peak at $d_a = 2^6$, and before the peak, the SVD entropy is noticeably larger than the ABB entropy. However, beyond the peak, the two entropies gradually converge and eventually coincide.

At $q = 2$, the entropies in average as the functions of subsystem size $\log_2 d_a$ are shown in Fig. 10. We observe the following phenomena:

- The pseudo entropy is enhanced as well, but its value is significantly smaller than that at $q = 4$.

- The modified pseudo entropy is suppressed and exhibits a plateau as well, but its value is significantly larger than that at $q = 4$.

- The SVD entropy and the ABB entropy exhibit a growth–peak–decline behavior as well. Both are smaller than their $q = 4$ counterparts, consistent with the observation that stronger chaos enhances eigenstate entanglement entropy in Hermitian SYK [116]. Moreover, the SVD entropy remains noticeably larger than the ABB entropy, indicating a non-flat singular-value spectrum.

By comparing the entropies from the nSYK at $q = 4$ and $q = 2$, we conclude that

- The enhancement of pseudo entropy is a common feature.

- The suppression and plateau of the modified pseudo entropy of transition matrices constructed in a bi-orthogonal basis are not exclusive for non-Hermitian chaotic systems; the same behaviors can arise even in a free non-Hermitian system.

- The SVD and ABB entropies, computed from bi-orthogonal eigenstates, are sensitive to non-Hermitian chaos, mirroring the sensitivity of eigenstate entanglement entropy to chaos in Hermitian systems.

It would be worthwhile to test these observations in other models in future work.

# 6 Close to the exceptional point

In this section, we are concerned with the behavior of different entropies when $\mathrm{Tr}[\tau] = \langle \psi_1 | \psi_2 \rangle$ goes to zero, which occurs in particular at the exceptional points of a $PT$-symmetric Hamiltonian. We demonstrate that the vanishing of the inner product leads to the divergence of the (modified) pseudo entropy, while the SVD entropy and ABB entropy remain finite.

We begin with a simple $PT$-symmetric Hamiltonian in a two-qubit system, which serves as an analog of the vectorized Lindbladian studied afterwards. We compute all four entropies for $\tau$, defined in the bi-orthogonal basis. Then we extend our analysis to the SYK Lindbladian.

## 6.1 Two-qubit system

To further illustrate the distinct behaviors of the above entropy measures, we first consider a simple two-qubit non-Hermitian model, in which two subsystems $a$ and $b$ are coupled as

$$H = iH_a + iH_b + \mu(\sigma_a^x \sigma_b^x + \sigma_a^z \sigma_b^z). \tag{68}$$

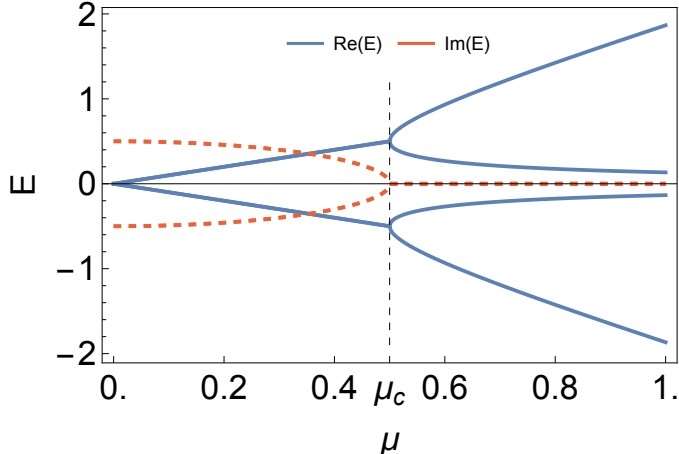

Figure 11: Spectrum of two-qubit model as the function of $\mu$.

where we choose local operators $H_a = \mathrm{diag}(-2/3, 1/3) \otimes \mathbb{I}$ and $H_b = \mathbb{I} \otimes \mathrm{diag}(1/6, 1/6)$ on the system $a$ and $b$, respectively. The third term describes the interaction between $a$ and $b$, with $\mu$ being a real positive coupling strength and $\sigma^{x,z}$ being Pauli matrices. This toy model shares the similar structure as the vectorized Lindbladian operator (74) discussed later.

We define the unitary operator as $P = \sigma^x \otimes \sigma^x$, and the anti-unitary operator $T$ as complex conjugation. With these definitions, the Hamiltonian $H$ in (68) satisfies $PT$ symmetry. Its eigenvalues are given by

$$E_{1,2} = \pm\mu - \frac{1}{2}\sqrt{4\mu^2 - 1}, \quad E_{3,4} = \pm\mu + \frac{1}{2}\sqrt{4\mu^2 - 1}, \tag{69}$$

as shown in Fig. 11. As $\mu$ decreases, each pair of eigenvalues approaches and eventually coalesces at the exceptional point $\mu_c = 1/2$. For $\mu < \mu_c$, the eigenvalues form complex-conjugate pairs. According to $PT$ symmetry properties of non-Hermitian Hamiltonians discussed in [121,122], the eigenvalues remain real for $\mu > \mu_c$, indicating that $PT$ symmetry of the eigenstates is preserved. For $\mu < \mu_c$, the spectrum becomes complex, signaling that $PT$ symmetry of the eigenstates is broken.

We construct the transition matrix $\tau$ from the bi-orthogonal eigenstates. For this model, $\tau$ has the same form across all four eigenstates, so we only need to consider one eigenstate. Two distinct expressions of $\hat{\tau}$ corresponding to $PT$-symmetric and $PT$-broken regions are obtained as

$$\hat{\tau}_{\mu>\mu_c} = \begin{pmatrix} \frac{1}{2} + \frac{i}{\sqrt{4\mu^2-1}} & 0 \\ 0 & \frac{1}{2} - \frac{i}{\sqrt{4\mu^2-1}} \end{pmatrix}, \quad \hat{\tau}_{0<\mu<\mu_c} = \begin{pmatrix} \frac{1}{2} + \frac{1}{\sqrt{1-4\mu^2}} & 0 \\ 0 & \frac{1}{2} - \frac{1}{\sqrt{1-4\mu^2}} \end{pmatrix}. \tag{70}$$

Accordingly, we can easily obtain $\bar{\tau} = \tilde{\tau}\tilde{\tau}^\dagger = \mathbb{I}/2$ in the $PT$-symmetric region. Furthermore, we compute the four entropies for both the $PT$-symmetric and $PT$-broken regions, and the results are displayed in Fig. 12, showing the entropy values as functions of $\mu$.

As seen in the left panel of Fig. 12, in the $PT$-symmetric region, both pseudo and modified pseudo entropy are real. For large $\mu$, they approach $\ln 2$. As $\mu$ decreases, the pseudo entropy grows and diverges positively at the exceptional point. However, the modified pseudo entropy decreases, becomes negative, and diverges negatively at $\mu_c$. In the $PT$-broken region, the modified pseudo entropy remains real and negative, while the

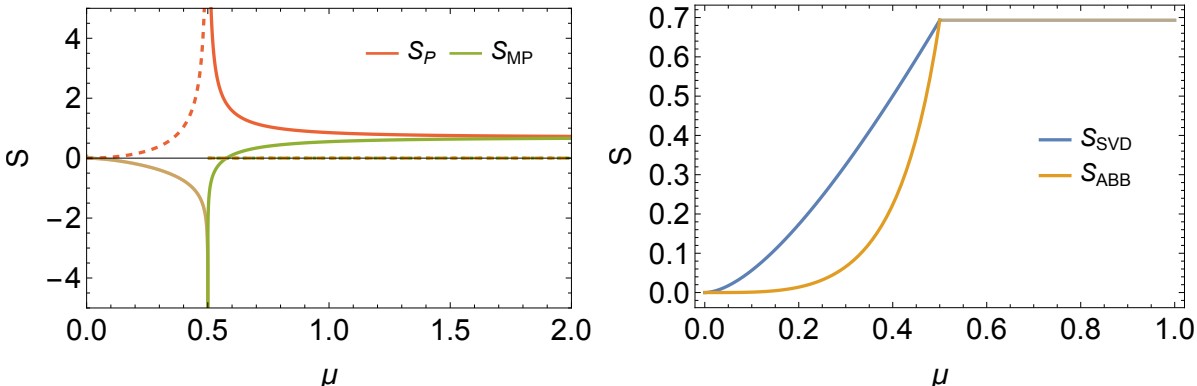

Figure 12: Left: The (modified) pseudo entropy of $\tau$. Solid and dashed lines denote real and imaginary parts, respectively. Their real parts coincide in the $PT$-broken region. Right: The corresponding SVD and ABB entropy of $\tau$.

pseudo entropy becomes complex. In the right panel, both SVD and ABB entropies take the real and positive values, and are strictly bounded by the logarithm of Hilbert space dimension in both the $PT$-symmetric and $PT$-broken regions. Notably, the SVD entropy is always larger than the ABB entropy in the $PT$-broken region.

We will see similar phenomena of entropies in SYK Lindbladian.

## 6.2 SYK Lindbladian

We now turn to the non-Hermitian vectorization of the Lindbladian superoperator in the SYK model.

In an isolated system without environmental interaction, the dynamics is governed by a Hermitian Hamiltonian, and $\rho$ evolves according to the Heisenberg evolution: $d\rho/dt = -i[H, \rho]$. However, a truly isolated system is an idealization. In practice, a system is always coupled, to some degree, with an external environment. Under the assumption of weak coupling and in the Markovian limit, the open system dynamics is described by the Lindblad master equation [123],

$$\frac{d\rho}{dt} = \mathcal{L}(\rho), \quad \mathcal{L}(\rho) = -i[H, \rho] + \sum_{\alpha} \left( L_{\alpha} \rho L_{\alpha}^{\dagger} - \frac{1}{2}\{L_{\alpha}^{\dagger} L_{\alpha}, \rho\} \right), \tag{71}$$

where $\mathcal{L}$ is the Lindbladian superoperator, describing the non-unitary evolution of the system, $H$ is the system's Hamiltonian, and $L_{\alpha}$ are jump operators encoding the interaction between the system and its environment.

The Lindbladian SYK models were proposed and investigated in [76, 124]. The SYK Hamiltonian incorporates the Gaussian-distributed random couplings [115],

$$H_{\text{SYK}} = i^{q/2} \sum_{1 \le i_1 < \cdots < i_q \le N} J_{i_1 \cdots i_q} \psi_{i_1} \cdots \psi_{i_q}, \tag{72}$$

where $\langle J_{i_1 \cdots i_q} \rangle = 0$, $\langle J_{i_1 \cdots i_q}^2 \rangle = \frac{J^2 (q-1)!}{N^{q-1}}$, and $\{\psi_i\}_{i=1}^{N}$ are Majorana fermionic operators satisfying $\{\psi_i, \psi_j\} = \delta_{ij}$. We choose the linear jump operators

$$L_i = \sqrt{\mu}\, \psi_i, \quad i = 1, 2, \cdots, N. \tag{73}$$

where $\mu \geq 0$ is the strength of dissipation. Specifically, we refer to the following vectorized Lindbladian as non-Hermitian: it is the operator obtained by mapping the Lindbladian superoperator onto the double-copy Hilbert space $\mathcal{H}_a \otimes \mathcal{H}_b$ via the CJ isomorphism [77, 78], and is given by [76]

$$\hat{\mathcal{L}} = -iH_a + i(-1)^{q/2}H_b - i\mu \sum_{i=1}^{N} \psi_{ai}\,\psi_{bi} - \mu\frac{N}{2}\mathbb{I}\,, \tag{74}$$

where $H_a = H_{\text{SYK}} \otimes \mathbb{I}$ and $H_b = \mathbb{I} \otimes H_{\text{SYK}}$, and $\psi_{ai}$, $\psi_{bi}$ act on the double-copy Hilbert space, obeying $\{\psi_{si}, \psi_{s'i'}\} = \delta_{ss'}\delta_{ii'}$, with $s, s' \in \{a, b\}$. The third term induced by the Lindblad terms describes the interaction between the systems $a$ and $b$, corresponding to the two subspaces of $\mathcal{H}_a \otimes \mathcal{H}_b$, respectively. The Lindblad equation in superoperator form (71) can be mapped to $d\,|\rho\rangle\,/dt = \hat{\mathcal{L}}\,|\rho\rangle$ with $|\rho\rangle$ being the vectorization of $\rho$.

In this work, we define the $PT$ symmetry of the vectorized Lindbladian in analogy with the definition used for non-Hermitian Hamiltonians in [121, 122], where the parity operator $P$ is a unitary transformation and the time reversal operator $T$ an anti-unitary one. The vectorized Lindbladian is $PT$-symmetric in the sense of

$$PT\hat{\mathcal{L}}(PT)^{-1} = \hat{\mathcal{L}}. \tag{75}$$

Further details of the vectorization and $PT$ symmetry can be found in Appendix. A. In this context, the $PT$ symmetry of an eigenstate is preserved if its associated eigenvalue is real. Conversely, if the eigenvalue is complex, then the $PT$ symmetry of the eigenstate is broken[4].

The spectrum of the vectorized Lindbladians (74) was computed numerically for finite $N$, revealing the complex eigenvalue distributions [76]. The real-time SYK Lindbladian dynamics was investigated in [129–132]. The symmetry classification of $PT$-symmetric SYK Lindbladians with more general Lindblad terms have also been investigated in [133, 134]. Various phenomena of non-Hermitian chaos have been studied in SYK Lindbladians, such as various form factors with dissipation [129, 135, 136] and dissipative scrambling [131, 132, 137, 138]. More recently, dissipative dynamics in bosonic SYK Lindbladian was studied in [139].

Here, we will numerically analyze the vectorized Lindbladian (74) and the $PT$ symmetry breaking behavior of eigenstates for varying $\mu$ at $N = 8$, $q = 4$. Furthermore, we investigate the behavior of the four entropy measures associated with $\tau$ constructed from the bi-orthogonal eigenstates, with particular focus on the behavior near exceptional points.

We numerically calculate the spectrum of $\hat{\mathcal{L}}$ for the varying parameter $\mu$ within one disorder realization, shown in the left panel of Fig. 13, where we have omitted the last constant term in (74) as it merely induces a shift of the real parts of the spectrum along the real axis. For sufficiently large $\mu$, the spectrum remains entirely real. As $\mu$ decreases, some pairs of eigenvalues approach each other, eventually colliding at some points, beyond

---

[4]However, the $PT$ symmetry employed here differs from that introduced in [125], where the symmetry is defined in the superoperator space. In that framework, the $PT$ transformation maps the Lindbladian $\mathcal{L} \rightarrow -\mathcal{L}$, leading to a distinct phase structure. For example, $PT$ symmetry breaking typically occurs in the small $\mu$ region, in contrast to the behavior observed here. There are also other definitions of the $PT$ symmetry with $P$ and $T$ defined in the operator space [126–128]. These framework also leads to the different phase structures. For example, in the $PT$-symmetric region, parts of the spectrum lie on the imaginary axis, while in the $PT$-broken region, all eigenvalues become real.

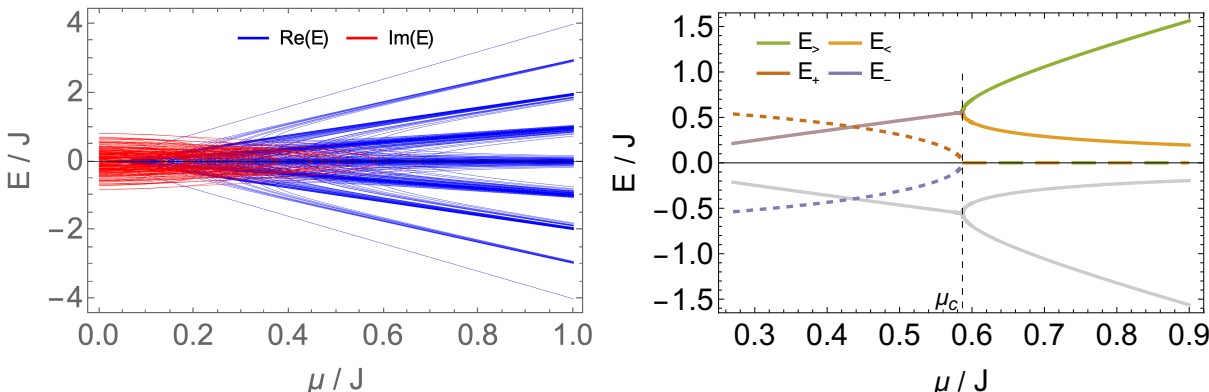

Figure 13: Left: Spectrum of the SYK Lindbladian for varying $\mu$. Blue and red lines denote the real and imaginary parts, respectively. Right: Evolution of two branches of representative eigenvalues. Solid and dashed lines indicate the real and imaginary parts, respectively. In the $PT$-symmetric region, we label the upper positive branch (green) by $E_>$ and the lower positive branch (orange) by $E_<$. The two branches coincide at the exceptional point $\mu_c/J = 0.586$ and become a complex-conjugate pair in the $PT$-broken region. We label the branches with positive (brown) and negative (purple) imaginary parts by $E_+$ and $E_-$, respectively. In addition, we include the two branches (gray) that are additive inverses of the colored branches; they have the same entropies as their inverse counterparts.

which they become complex conjugate to each other. We trace two representative branches of eigenvalues that first undergo their transitions from $\{E_>, E_<\}$-branches to $\{E_+, E_-\}$-branches as $\mu$ decreases, as shown in the right panel of Fig. 13, where those branches of eigenvalues obey

$$
\begin{aligned}
\mu > \mu_c : & \quad E_>, E_< \in \mathbb{R}, \quad E_> > E_<; \\
\mu = \mu_c : & \quad E_< = E_> = E_+ = E_-; \\
\mu < \mu_c : & \quad E_+, E_- \in \mathbb{C}, \quad E_+^* = E_-.
\end{aligned}
\tag{76}
$$

Each eigenvalue corresponds to a left-right eigenstate pair in the bi-orthogonal basis. When $\mu > \mu_c$, the two corresponding eigenstates $\{|R_>\rangle, |R_<\rangle\}$ respect $PT$ symmetry. At $\mu = \mu_c$, they coincide. When $\mu < \mu_c$, two new eigenstates $\{|R_+\rangle, |R_-\rangle\}$ are generated and do not respect $PT$-symmetry. So, for these two branches, we call $(\mu_c, \infty)$ the $PT$-symmetric region, $\mu_c$ the exceptional point, and $[0, \mu_c)$ the $PT$-broken region[5]

We then construct the transition matrices $\tau = \mathrm{Tr}_b |R\rangle \langle L|$ by tracing out the system $b$ for each of right eigenstates $\{|R_>\rangle, |R_>\rangle\}$ or $\{|R_+\rangle, |R_-\rangle\}$ and their left counterparts. For each transition matrix in $\{\tau_<, \tau_>\}$ or $\{\tau_+, \tau_-\}$, we compute the (modified) pseudo entropy, the SVD entropy, and the ABB entropy.

The (modified) pseudo entropy is shown in Fig. 14. In the $PT$-symmetric region ($\mu > \mu_c$), the two branches of the pseudo entropy remain real and positive. As $\mu$ decreases, they increase and ultimately diverge at the exceptional point $\mu_c$. The modified pseudo entropy for the branch $E_<$ decreases as $\mu$ decreases, eventually becomes negative, and diverges to $-\infty$ at $\mu_c$, whereas for the branch $E_>$, it exhibits non-monotonic behavior and features a sharp increase in the vicinity of $\mu_c$. In the $PT$-broken region ($\mu < \mu_c$), all

---

[5]Here, we define $PT$ symmetry at the level of individual eigenstates, in contrast to the criterion in [121, 122], where a Hamiltonian is deemed $PT$-symmetric only if its entire spectrum is real.

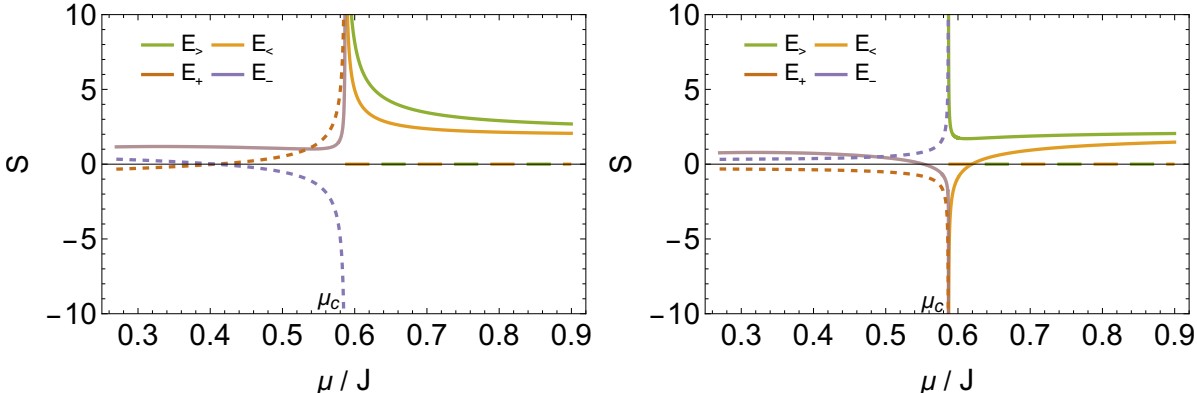

Figure 14: The pseudo entropy (left) and modified pseudo entropy (right) of $\tau$ constructed from the bi-orthogonal eigenstates along the two branches in Fig. 13.

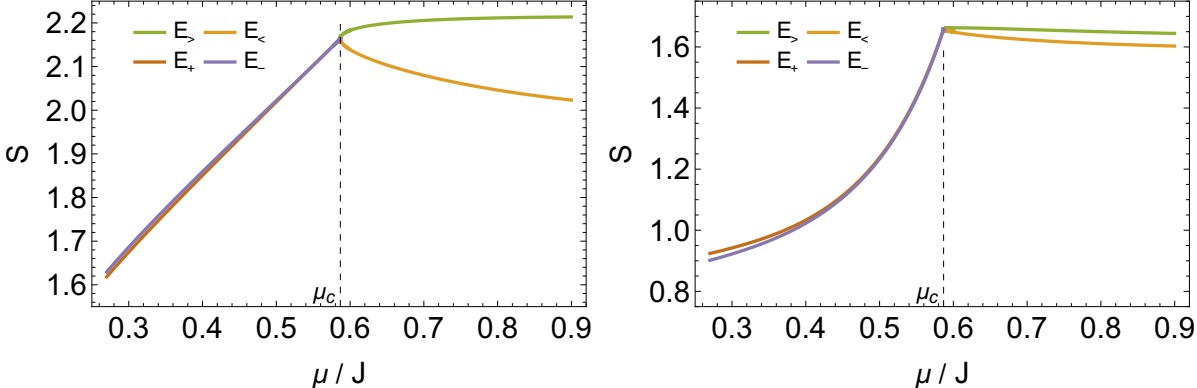

Figure 15: The SVD entropy (left) and ABB entropy (right) of the same $\tau$.

(modified) pseudo entropies become complex and diverge at $\mu_c$. Both branches $E_+$ and $E_-$ share the same real part of the (modified) pseudo entropy, and their imaginary parts are additive inverses of each other.

The SVD and ABB entropies are shown in Fig. 15. Unlike the (modified) pseudo entropy, they remain real, positive, and strictly bounded by $\ln d_a = \frac{N}{2}\ln 2$. In the $PT$-symmetric region, the branch $E_>$ exhibits larger SVD and ABB entropies than the branch $E_<$. At $\mu_c$, the entropies of each degenerate pair coincide and then decrease as $\mu$ is lowered further into the $PT$-broken region. These decreases are consistent with the tendency of the two eigenstates $\{|R_>\rangle, |R_<\rangle\}$ of the double-copy system to disentangle as the coupling $\mu$ is reduced. We also observe that the two branches take distinct values and that the SVD entropy is consistently greater than the ABB entropy for all $\mu$.

Thus, we have shown that the divergence of the (modified) pseudo entropy challenges the claim that these entropies quantify quantum entanglement, even when they take real values. By contrast, both the SVD entropy and the ABB entropy behave consistently with the usual entanglement entropy of a single quantum state and align with the tendency toward disentanglement.

Finally, we observe some opposite monotonic behaviors of entropies, which also challenge the claim that (modified) pseudo entropy quantify quantum entanglement. For example, in the $PT$-symmetric region of the SYK Lindbladian, although all the four entropies of the $E_<$ branch are real, the monotonicity of the modified pseudo entropy as a function of $\mu$ is opposite to the monotonicity of others entropies. Similar situation happens in the $PT$-symmetric region of the two qubits system, where both the pseudo

entropy and modified pseudo entropy change dramatically for varying $\mu$ but the SVD entropy and ABB entropy stay constant.

# 7 Conclusion and outlook

## 7.1 Conclusion

In this paper, we explored the transition matrix $\tau$ that describes a post-selection process transferring part of the information of a quantum state, either pure or mixed, from one side to another. In contrast to teleportation protocols, the transition matrix is not a trace-preserving map, due to post-selection. Hence, we employ entropy-based measures to quantify the amount of information transferred.

We introduced the ABB entropy of the transition matrix to quantify information transfer, which could be interpreted as the relative entropy between the input maximally mixed state and its output state bridged by the transition matrix (Sec. 2.2). The ABB entropy also avoids the issues encountered in the (modified) pseudo entropy and the SVD entropy. The pseudo entropy suffers from ambiguity due to the multi-valued logarithm. Both the pseudo and modified pseudo entropies can diverge or take complex values. Although the SVD entropy circumvents these problems, it lacks a clear probabilistic interpretation based on entanglement distillation of the two quantum states that construct the transition matrix (Sec. 3).

Subsequently, we demonstrated that the ABB entropy of the transition matrix does not increase under the addition of measurements and non-unitary operations, as illustrated in Fig. 3, in agreement with the behavior of conventional entanglement entropy for pure quantum states under LOCC [71]. By contrast, the (modified) pseudo entropy and the SVD entropy do not necessarily exhibit this monotonicity.

We showed that the ABB entropy of the large-copy transition matrices can be concentrated into that of the sub transition matrix with the highest probability (Sec. 3.2.1), following the notion of entanglement distillation in [69]. That probability corresponds to the highest probability of generalized measurement on a mixed state. We also examined the probabilistic interpretation of the pseudo and SVD entropies of the transition matrix in [16, 67]. We found that a meaningful concentration interpretation for the (modified) pseudo entropy is possible only when the normalized transition matrix has a real and non-negative spectrum, allowing them to be associated with a sub transition matrix. However, even in this case, the so-called "probability" of the concentration does not correspond to a true measurement probability in the quantum mechanical sense. For the transition matrices with negative or complex eigenvalues, it is even not possible to construct any probability-like distribution over the sub-transition matrices (Sec. 3.2.2). Similarly, for the SVD entropy, although the singular spectrum of the transition matrix is real and non-negative, the associated distribution over sub transition matrices still does not represent a genuine probability distribution in a quantum measurement process.

We computed the ensemble averages of the four entropies for transition matrices constructed from two independent Haar random states in Sec. 4, and from biorthogonal eigenstates of non-Hermitian random systems, including the GinUE and the non-Hermitian SYK model, in Sec. 5. In all cases, the SVD and ABB entropies exhibit behavior similar to the subsystem entanglement entropy of a single random state at large system size. By contrast, the pseudo entropy exceeds the constraint from the subsystem size and the modified pseudo entropy does not scale with subsystem size.

The contradistinction between these entropies becomes especially pronounced in a $PT$-symmetric model, as discussed in Sec. 6.2. Both the pseudo entropy and the modified pseudo entropy exhibit divergences near exceptional points and acquire complex values in the $PT$-broken region, while the SVD and ABB entropies remain finite in both the $PT$-symmetric and $PT$-broken regions, and capture the entanglement properties of eigenstates.

## 7.2   Outlook

For two independent Haar random states, the subsystem-size asymmetry of the SVD and ABB entropies seen in Fig. 5 plausibly originates from subleading contributions in (62), so a detailed evaluation of these terms is warranted in future work. Moreover, while we numerically observe a plateau in the modified pseudo entropy in this case and attribute the emergence of negative values to spectral properties, an analytical computation of the spectrum for the normalized transition matrix remains for future investigation.

For bi-orthogonal states in the GinUE setting, although the SVD and ABB entropies exhibit Page-curve-like behavior in numerics, analytic expressions for them are still lacking, unlike the modified pseudo entropy, whose plateau is predictable [63]. Estimating their scaling behavior in future work would support a more compelling universal statement.

The ABB entropy has seen limited investigation in field-theoretic settings, and its structural properties remain largely unexplored. By contrast, pseudo entropy has been thoroughly studied in free scalar field theories and spin models [22, 23], where it displays area-law behavior, saturation, and the so-called non-positivity difference. In the biorthogonal basis of $PT$-symmetric systems, the pseudo and modified pseudo entropies exhibit logarithmic scaling with negative central charges at the critical points of the non-Hermitian spin and SSH models [50, 56, 60]. It remains an interesting question whether analogous behavior holds for the SVD and ABB entropies in such models.

Investigations of the ABB entropy within the framework of gravity duals are still lacking. For pseudo entropy, the corresponding transition matrix was realized in holographic framework [16, 36, 37]. A promising direction is to construct the gravity dual of the corresponding transition matrix for the ABB entropy in this setting.

Pseudo entropy is known to become complex in certain time-like configurations [39, 140], complicating its geometric interpretation [141]. In contrast, the SVD and ABB entropies remain real and positive even in time-evolved or non-unitary contexts, making it a potentially more robust alternative. Their construction also avoids ambiguities related to analytic continuation and partial swaps in imaginary time. Systematic exploration of its behavior under both unitary and non-unitary dynamics, could help establish SVD or ABB entropy as a potential diagnostic tool for temporal entanglement [142].

# Acknowledgments

We are grateful to Zixia Wei, Giuseppe Di Giulio, and Jie Ping Zheng for helpful discussions. We are especially grateful to Zixia Wei for his detailed explanations and insightful suggestions on pseudo and SVD entropy. We are also indebted to Jonah Kudler-Flam for explanations of the numerical computations in the GinUE. R.M. and Z.-Y.X. acknowledge funding by DFG through the Collaborative Research Center SFB 1170 ToCoTronics, Project-ID 258499086-SFB 1170, as well as Germany's Excellence Strategy through the

Würzburg-Dresden Cluster of Excellence on Complexity and Topology in Quantum Matter - ct.qmat (EXC 2147, project-id 390858490). Z.C. is financially supported by the China Scholarship Council. Z.-Y.X. also acknowledges support from the Berlin Quantum Initiative.

# A  Vectorization and PT-symmetry of the SYK Lindbladian

Here we introduce the vectorization of the SYK Lindbladian via the CJ isomorphism and analyze its $PT$-symmetry, following [76, 130, 143].

The Hilbert space $\mathcal{H}$ of $N$ Majorana fermions is in dimension $2^{N/2}$. We use the Jordan-Wigner transformation [144, 145] to yield the $(2^{N/2} \times 2^{N/2})$-dimensional matrix representation of the $N$ Majorana fermion,

$$
\begin{aligned}
\psi_{2k-1} &= \frac{(-1)^{k-1}}{\sqrt{2}} \, \sigma_z^{\otimes(k-1)} \otimes \sigma_x \otimes \mathbb{I}^{\otimes(N/2-k)}, \\
\psi_{2k} &= \frac{(-1)^{k-1}}{\sqrt{2}} \, \sigma_z^{\otimes(k-1)} \otimes \sigma_y \otimes \mathbb{I}^{\otimes(N/2-k)}.
\end{aligned}
\tag{77}
$$

For later convenience, we define a Hermitian chiral matrix $S = i^{N(N-1)/2} \prod_{i=1}^{N} \sqrt{2} \, \psi_i$, where the product of Majorana fermions is ordered sequentially from $i = 1$ to $N$, and $S$ satisfies $\{S, \psi_i\} = 0$ and $S^2 = 1$.

To implement the CJ isomorphism, we introduce the $2^N$-dimensional double-copy Hilbert space $\mathcal{H} \otimes \mathcal{H}$. We define $2N$ Majorana fermion operators $\{\psi_{ai}, \psi_{bi}\}_{i=1}^{N}$ acting on the double-copy Hilbert space as

$$
\psi_{ai} = \psi_i \otimes S, \quad \psi_{bi} = \mathbb{I} \otimes \psi_i, \quad i = 1, \cdots, N,
\tag{78}
$$

which obey the anti-commutation relation $\{\psi_{si}, \psi_{s'i'}\} = \delta_{ss'}\delta_{ii'}$ with $s, s' \in \{a, b\}$.

To define a CJ isomorphism, we uniquely specify a MES

$$
\mathbb{I} \rightarrow |0\rangle = \frac{1}{2^{N/4}} \sum_{j=1}^{2^{N/2}} |j\rangle \otimes |\tilde{j}\rangle \quad \text{with} \quad \langle 0|0\rangle = 1,
\tag{79}
$$

in the double-copy Hilbert space, by imposing the requirement

$$
\psi_{ai} |0\rangle = -i\psi_{bi} |0\rangle, \quad \forall i.
\tag{80}
$$

The detailed construction of $|0\rangle$ can be seen in Appendix A of [130]. Then, the CJ isomorphism, as a map from the operator space on the single-copy Hilbert space to the double-copy Hilbert space, is defined as

$$
O \rightarrow O \otimes \mathbb{I} |0\rangle, \quad \forall O.
\tag{81}
$$

Using this CJ isomorphism and denoting the image of the density matrix $\rho$ as $|\rho\rangle =$

$\rho \otimes \mathbb{I} \, |0\rangle$, we can demonstrate $\mathcal{L}(\rho) \rightarrow \hat{\mathcal{L}} \, |\rho\rangle$ with the vectorization (74) via

$$H \rho \rightarrow H \rho \otimes \mathbb{I} \, |0\rangle = i^{q/2} J_{i_1 \cdots i_q} (\psi_{i_1} \otimes \mathbb{I}) \cdots (\psi_{i_q} \otimes \mathbb{I})(\rho \otimes \mathbb{I}) \, |0\rangle \tag{82}$$

$$= i^{q/2} J_{i_1 \cdots i_q} (\psi_{i_1} \otimes S) \cdots (\psi_{i_q} \otimes S) \, |\rho\rangle = i^{q/2} J_{i_1 \cdots i_q} \psi_{ai_1} \cdots \psi_{ai_q} \, |\rho\rangle = H_a \, |\rho\rangle \, ,$$

$$\rho H \rightarrow \rho H \otimes \mathbb{I} \, |0\rangle = i^{q/2} J_{i_1 \cdots i_q} (\rho \otimes \mathbb{I})(\psi_{i_1} \otimes S) \cdots (\psi_{i_q} \otimes S) \, |0\rangle \tag{83}$$

$$= i^{q/2} J_{i_1 \cdots i_q} (\rho \otimes \mathbb{I})(-i)^q (\mathbb{I} \otimes \psi_{i_1}) \cdots (\mathbb{I} \otimes \psi_{i_q}) \, |0\rangle = (-1)^{q/2} H_b \, |\rho\rangle \, ,$$

$$\psi_i \rho \psi_i \rightarrow \psi_i \rho \psi_i \otimes \mathbb{I} \, |0\rangle = (\psi_i \otimes S)(\rho \otimes \mathbb{I})(\psi_i \otimes S) \, |0\rangle = -i \psi_{ai} \psi_{bi} \, |\rho\rangle \, , \tag{84}$$

$$\frac{1}{2} \left( \psi_i^\dagger \psi_i \rho + \rho \psi_i^\dagger \psi_i \right) \rightarrow \frac{N}{4} \left( \mathbb{I}\rho + \rho\mathbb{I} \right) \otimes \mathbb{I} \, |0\rangle = \frac{N}{2} \mathbb{I} \otimes \mathbb{I} \, |\rho\rangle \, , \tag{85}$$

where we use the Einstein summation convention, and $H_a$ and $H_b$ are both SYK Hamiltonians with identical random couplings, as given in (72). One can also easily check $\hat{\mathcal{L}} \, |0\rangle = 0$.

Obviously, $\hat{\mathcal{L}}$ is non-Hermitian, i.e., $\hat{\mathcal{L}}^\dagger \neq \hat{\mathcal{L}}$, but it enjoys $PT$ symmetry [134],

$$\hat{\mathcal{L}} = PT\hat{\mathcal{L}}(PT)^{-1}, \quad P = \exp\left( -i\frac{\pi}{2} \sum_{i=1}^{N} \psi_{ai} S_a \psi_{bi} \right), \quad T = QK, \tag{86}$$

where $S_a = S \otimes \mathbb{I}$, $Q = \prod_{i=1} \sqrt{2} \, \psi_{a(2i)} \prod_{k=1} \sqrt{2} \, \psi_{b(2k)}$, and $K$ is complex conjugation in the matrix representation (77). The unitary operator $P$ acts as a parity operator, while the anti-unitary operator $T$ represents time reversal. It can be easily checked that $PP^\dagger = 1$, $T^2 = (-1)^{N/2}$. Using the anti-commutation relations, the operator $P$ can be further simplified as

$$P = \prod_{i=1}^{N} \frac{1}{\sqrt{2}} \left( 1 - 2i\psi_{ai} S_a \psi_{bi} \right), \tag{87}$$

where the product is still ordered sequentially from $i = 1$ to $N$. Under the action of $P$, the transformations of Majorana fermions are given by

$$P\psi_{ai}P^{-1} = -iS_a\psi_{bi}, \quad P\psi_{bi}P^{-1} = -i\psi_{ai}S_a. \tag{88}$$

Similarly, the action of $T$ is given by

$$TiT^{-1} = -i, \quad T\psi_{ai}T^{-1} = \psi_{ai}, \quad T\psi_{bi}T^{-1} = \psi_{bi}. \tag{89}$$

One can directly verify (86) via the above properties.

However, the eigenstates of $\hat{\mathcal{L}}$ do not necessarily respect $PT$ symmetry unless the corresponding eigenvalues are real. This follows from

$$\hat{\mathcal{L}} \, |R\rangle = E \, |R\rangle \Rightarrow \hat{\mathcal{L}} PT \, |R\rangle = PT\hat{\mathcal{L}}(PT)^{-1}PT \, |R\rangle = PT\hat{\mathcal{L}} \, |R\rangle = E^* PT \, |R\rangle \, . \tag{90}$$

Thus, if the spectrum is real, the eigenstates remain invariant under the $PT$ transformation, which means that they satisfy the $PT$ symmetry. Conversely, if the spectrum contains complex eigenvalues, the $PT$ transformation just maps an eigenstate $|R\rangle$ to another state $PT \, |R\rangle$ with eigenvalue $E^*$. This establishes that the spectrum of $\hat{\mathcal{L}}$ is symmetric under complex conjugation.

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
