# Peer review of "Entropy Measures for Transition Matrices in Random Systems"

_SciPost Physics_

## Round 2 · Referee Report · Anonymous (Referee 2) · 2025-11-13

The referee discloses that the following generative AI tools have been used in the preparation of this report:
gramma check for the comments
Strengths
-
A comparison of different entropic measures for transition matrices reveals that the ABB entropy possesses a clear probabilistic interpretation, unlike other entropies.
-
Several examples are provided to illustrate these different entropic measures.
Weaknesses
-
Lack of heuristics understanding of asymmetric entropic shapes of ABB and SVD entropies in the first three examples.
-
Several claims are not justified and are not convincing.
Report
Their results are interesting and could lead to future investigations of ABB entropy. However, the provided examples show that all the entropies exhibit drastic changes near the EPs and all can detect quantum chaotic properties. This suggests that the ABB entropy does not stand out from the others, but is simply another measure that fits the standard quantum information perspective.
I think the paper is interesting enough to be published in SciPost Physics after some revisions.
Requested changes
-
In Table 1, the normalizations for \hat{\tau},\bar{\tau} are based on the trace norm. while the \tilde{\tau} is based on the post-selected maximally mixed state. The authors should clarify this.
-
The main reason the ABB entropy has a proper probability interpretation appears to be that it is just the entropy measure of the post-selected maximally entangled states under the transition matrix, rather than of the transition matrix itself. A crucial question is : if one chose a state other than maximally entangled state, would this property change?
-
The n=1/2 Renyi ABB entropy is equal to twice the SVD entropy. Is it correct?
-
On Page 12, the authors discuss the SVD entropy fails to be a Schur-concave function and also point out that the (modified ) pseudo entropy coincides with the SVD entropy when the transition matrix is Hermitian. I wonder: if the modified pseudo entropy is real but the transition matrix is non-Hermitian, i.e., eigenvalues are real but can be negative, can the authors numerically demonstrate the failure of Schur-concave function along the majorization path?
-
For Figs. 5, 7, 9, 10, is there any heuristic understanding of why the ABB and SVD entropies are asymmetric?
-
For Fig.12, the SVD and ABB entropies are flat in the PT symmetric region. Why do these entropies not vary as a function of \mu? Is \tau \tau^\dagger independent of \mu in the PT-symmetric region?
-
For Fig.12, in the PT-broken region \mu \in [0,0.5], do the real parts of pseudo and modified pseudo entropies coincide? The colors in Fig.12 appear to overlap.
-
In the non-Hermitian SYK example, the authors claim that the SVD and ABB entropies are sensitive to non-Hermitian chaos. Their reasoning is that q=2 has smaller value of than q=4. I don't think it is a strong evidence of sensitivity to non-Hermitian chaos. Am I missing something?
-
Comparing Figs. 7 and 9, the SVD and ABB entropies for both examples look almost the same. Are they the same?
-
All the examples show that the ABB entropy is bounded by SVD entropy. Is it accidental, or can one prove this inequality, AAB<=SVD?
Recommendation
Ask for minor revision
Strengths
- They give explicit results for several models
Weaknesses
- The review of the entropies is confusing
Report
In the first half of this paper, which is mostly a review, the authors discuss four different types of entanglement entropy. Several examples are worked out in the second half of the paper. What I miss in this paper is the physical motivation for introducing these enetropies. Why do we need them? The lack of motivation makes reading the first half confusing. The worked out examples are in the second half of the paper, but it is not clear exactly what is calculated. In order to reproduce the figures, I have to guess what the authors are doing, snd it should not be like that. The authors should be very clear about the equation they have been using to calculate the entropies.
Despite of this criticism I think that the paper after the authors have addressed the following issues:
-
It would be a good idea to shorten the first half of the paper. What should be clear are the general properties that a transition matrix should satisfies. From reading the paper I get the idea that it can be anything. The connection the authors make with teleporttion is confusing because the examples are not at all related to teleportatiion.
-
Eq. (6): $S[\rho || \sigma )$ is not defined. Pleas give an explicit definition.
-
Line above eq. (19): we do not need “novel”
-
It is not clear what the authors mean by distillation and why it is useful. Please explain and motivate.
-
Explain Eq. (36). I assume that $d_{1\vec k^*}$ is still given by the bynomial factor. Mention at least Stirling. What is exaclyt $|0\rangle $ in (54). Can the author add an appendix with an expression for (54) with all indices written out explicitly including the tensor products.
-
Eq. (55): Du and dV do not have indices but U and V do. Are they the same?
-
Fi.g 5:: Why some curves are symmetric and other are not. Please explain exactly which equations have been used for these figures.
-
What is the transition matrix for which the eigenvalues in Fig. 6 have been calculated?
-
a and b are the left and right eigenvectors of the Ginibre ensemble according to (64) and must have the same dimension, but this does not square with Fig. 7. Please explain exactly what is calculated and which equations have been used. Also explain why some curves appear symmetric and others do not.
-
I have the same questions for Fig. 7.
-
3 Lines below Fig. 7: It is not class A of the AZ classification but class A of the Bernars-LeClair classiufication of non hermitian Random matrices which was completed by Ueda and collaborators.
-
What is the transition matrix used for Fig. 8? Give explicit equations.
-
4 lines below (67\: change AZ class to Bernard-LeClair class.
-
Could the authors explain exactly which equations are used to obtain Figs. 9-10? It should be done in such a way that the figures can be reproduced without any guess work.
-
It is not clear which eqs. are used for Fig. 10. Please provied all details for the transition operator and how it is obtained from the nSYK.
-
For the curves in Figs. 14 and 15, do the authors only consider two states or more than two. The exceptional point probably depends on the realization.. Is there only one realization? Please state explicitly in the. caption.
Requested changes
See Report
Recommendation
Ask for major revision

---

## Editorial Decision

unknown